# Interactions between interfaces dictate stimuli-responsive emulsion behaviour

Marcel Rey [1,2,7] ✉, Jannis Kolker[3,7], James A. Richards [1], Isha Malhotra[3], Thomas S. Glen[1], N. Y. Denise Li [1], Fraser H. J. Laidlaw [1], Damian Renggli [4], Jan Vermant [4], Andrew B. Schofield[1], Syuji Fujii [5,6], Hartmut Löwen[3] & Paul S. Clegg [1]

Stimuli-responsive emulsions offer a dual advantage, combining long-term storage with controlled release triggered by external cues such as pH or temperature changes. This study establishes that thermo-responsive emulsion behaviour is primarily determined by interactions between, rather than within, interfaces. Consequently, the stability of these emulsions is intricately tied to the nature of the stabilizing microgel particles - whether they are more polymeric or colloidal, and the morphology they assume at the liquid interface. The colloidal properties of the microgels provide the foundation for the long-term stability of Pickering emulsions. However, limited deformability can lead to non-responsive emulsions. Conversely, the polymeric properties of the microgels enable them to spread and flatten at the liquid interface, enabling stimuli-responsive behaviour. Furthermore, microgels shared between two emulsion droplets in flocculated emulsions facilitate stimuli-responsiveness, regardless of their internal architecture. This underscores the pivotal role of microgel morphology and the forces they exert on liquid interfaces in the control and design of stimuli-responsive emulsions and interfaces.

Pickering emulsions are metastable dispersions of two immiscible liquids, kinetically stabilized by colloidal particles that partially wet both fluids[1,2]. Although discovered more than a century ago, they received renewed interest due to the desire to decrease the use of potentially environmentally damaging surfactants[3–5] and the increased abundance of particles able to adsorb at an oil–water interface[6,7].

With particles strongly adsorbed to the interface, Pickering emulsions exhibit long-term stability, which makes them ideal for storage. However, many applications, such as in biomedicine[8] or catalysis[3,9–11] also require the release of the emulsified liquid. Thus, recent work has focussed on controlled release upon external stimuli[12]. E.g., this can occur through modifying the continuous phase via a change of pH[13–17], the addition of sugar[18] or oxidizing agents[19], the bubbling of $CO_2$[20], addition of solvents[21,22], or via external triggers including light[11] and temperature[23–26]. Frequently used stabilizers in thermo-responsive emulsions are poly(N-isopropylacrylamide) (PNIPAM) microgels particles, which transition from a swollen to a collapsed state above their volume phase transition temperature ($T_{VPT}$) of 32 °C. As a result, emulsions stabilized by PNIPAM microgels are stable at room temperature, but can destabilize above $T_{VPT}$[24–32].

The fundamental mechanism behind the rupture of such oil in water emulsions is, 15 years after its discovery[30,31], still under debate[28,29]. Earlier reports attributed the breaking directly to the volume phase transition of the stabilizing microgels. It was

[1]School of Physics and Astronomy, The University of Edinburgh, Peter Guthrie Tait Road, Edinburgh EH9 3FD, UK. [2]Department of Physics, University of Gothenburg, SE-41296 Gothenburg, Sweden. [3]Institute for Theoretical Physics II: Soft Matter, Heinrich-Heine University Düsseldorf, D-40225 Düsseldorf, Germany. [4]Department of Materials, ETH Zürich, Vladimir-Prelog-Weg 5, 8093 Zürich, Switzerland. [5]Department of Applied Chemistry, Faculty of Engineering, Osaka Institute of Technology, 5-16-1 Omiya, Asahi-ku, Osaka 535-8585, Japan. [6]Nanomaterials Microdevices Research Center, Osaka Institute of Technology, 5-16-1 Omiya, Asahi-ku, Osaka 535-8585, Japan. [7]These authors contributed equally: Marcel Rey, Jannis Kolker. ✉e-mail: marcel.rey@physics.gu.se

speculated that when heated above $T_{VPT}$, the stabilizing microgels shrink laterally and the reduced interfacial coverage destabilizes the emulsions[29–31,33–37] with a potential change in the mechanical properties of the interfacial microgel monolayer[38–41]. It was also proposed that microgels desorbed from the oil/water interface, again lowering coverage[30,36], although this did not appear in all reported scenarios[38,39,42].

Two recent studies of microgel monolayers showed that they persisted throughout temperature cycling and that no desorption occurred[43,44]. Additionally, the lateral dimensions of the microgels did not change upon heating and, therefore, there were no changes in the interfacial assembly[43,44]. Ellipsometry[44], neutron reflectometry[45] and molecular dynamic simulations[43,45,46] suggest that only the part of the microgel exposed to the water phase changes with temperature and collapses. With these insights, the previously established direct destabilisation mechanism is brought into question[28,29]. However, to enable the rational design of responsive emulsions, the origin of the destabilisation cannot remain a mystery and must be understood.

In this work, we first establish, using interfacial shear rheology, that thermo-responsive behaviour is not due to the lateral assembly. We then use cryogenic scanning electron microscopy and monomer-resolved Brownian dynamics simulations to reveal that the macroscopic emulsion stability is instead linked to the individual microgel morphologies and the forces they exert on the liquid interfaces. Finally, we investigate a distinct series of core-shell structured microgels to establish design criteria in responsive emulsions.

## Results

### Model interfaces reveal that the destabilisation mechanism is unrelated to lateral microgel properties

To address the microscopic origin of thermo-responsive emulsions, we probe monolayers of PNIPAM microgels with varying crosslinking densities and architectures (Supplementary Fig. 1a–c) at a dodecane/water interface using oscillatory interfacial shear rheology with increasing temperature. Previously proposed mechanisms for stimuli-responsive destabilisation would lead to either a fluidised interface ($G^{s\prime}$ « $G^{s\prime\prime}$) due to lower surface coverage (Fig. 1a (i)-(ii)), comparable to bulk fluidisation with reduced volume fraction[47], or a weaker interface due to aggregation (Fig. 1a (iii)). With strong changes in the lateral microgel interactions, the surface storage modulus ($G^{s\prime}$, solid-like) should drop below the surface loss modulus ($G^{s\prime\prime}$, liquid-like). Using a double wall ring geometry (Fig. 1b)[48], the interface is characterised by a low-frequency strain amplitude ($\gamma_O$) sweep at temperature $T = 21\,°C$. The linear viscoelastic properties are monitored, while $T$ is increased above $T_{VPT}$, before a $\gamma_O$ sweep at 43 °C.

Ultra-low crosslinked (ULC) microgels create only a weakly elastic interface (Fig. 1c (blue)), with $G^{s\prime} = 1.8(1) \times 10^{-4}\,Pa\,m$ (filled) above $G^{s\prime\prime}$ (open) in the linear regime from the resolution limit (shading) to $\gamma_O = 0.1$. With further increasing strain amplitude, $G^{s\prime}$ decreases while $G^{s\prime\prime}$ rises, until crossing, an operative definition of yielding from solid to liquid-like. Upon decreasing strain amplitude, the interface returns to $G^{s\prime} = 1.5(1) \times 10^{-4}\,Pa\,m$, and solid behaviour is recovered (Supplementary Fig. 2). At $\gamma_O = 0.05$, with increasing temperature $G^{s\prime}$ remains above

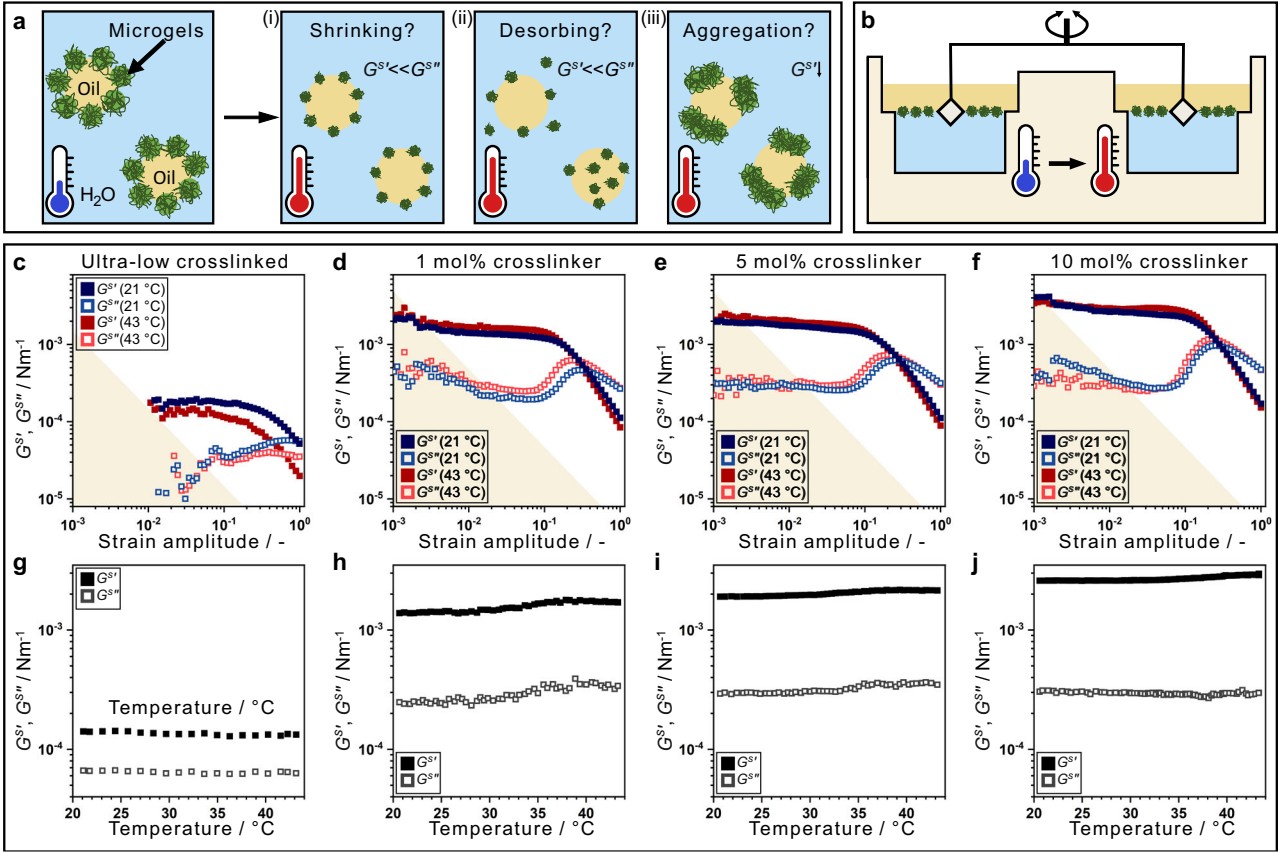

**Fig. 1 | Interfacial response of thermo-responsive PNIPAM microgels.**
**a** Previously proposed destabilisation mechanisms: covered interface to fluidised interface with lower coverage due to (i) shrinkage, (ii) desorption, or (iii) weakening of the interface due to aggregation. **b** Interfacial shear rheology with double wall ring geometry. **c**–**j** Interfacial rheological response to changing temperature: **c**–**f** Oscillatory strain amplitude sweep for (**c**) ultra-low crosslinked (ULC) microgels at a frequency $f = 0.1\,Hz$, microgels with (**d**) 1 mol% crosslinker, (**e**) 5 mol% crosslinker and (**f**) 10 mol% crosslinker at $f = 0.2\,Hz$. Storage ($G^{s\prime}$, filled) and loss ($G^{s\prime\prime}$, open) moduli with strain amplitude, $\gamma_O$, at low temperature, $T < T_{VPT}$ (blue), and high temperature, $T > T_{VPT}$ (red). The shading indicates the resolution limit[81]. **g**–**j** Linear viscoelastic response with increasing $T$: (**g**) ULC microgels at $\gamma_O = 0.05$, (**h**–**j**) microgels with (**h**) 1 mol% crosslinker, (**i**) 5 mol% crosslinker and (**j**) 10 mol% crosslinker each at $\gamma_O = 0.01$.

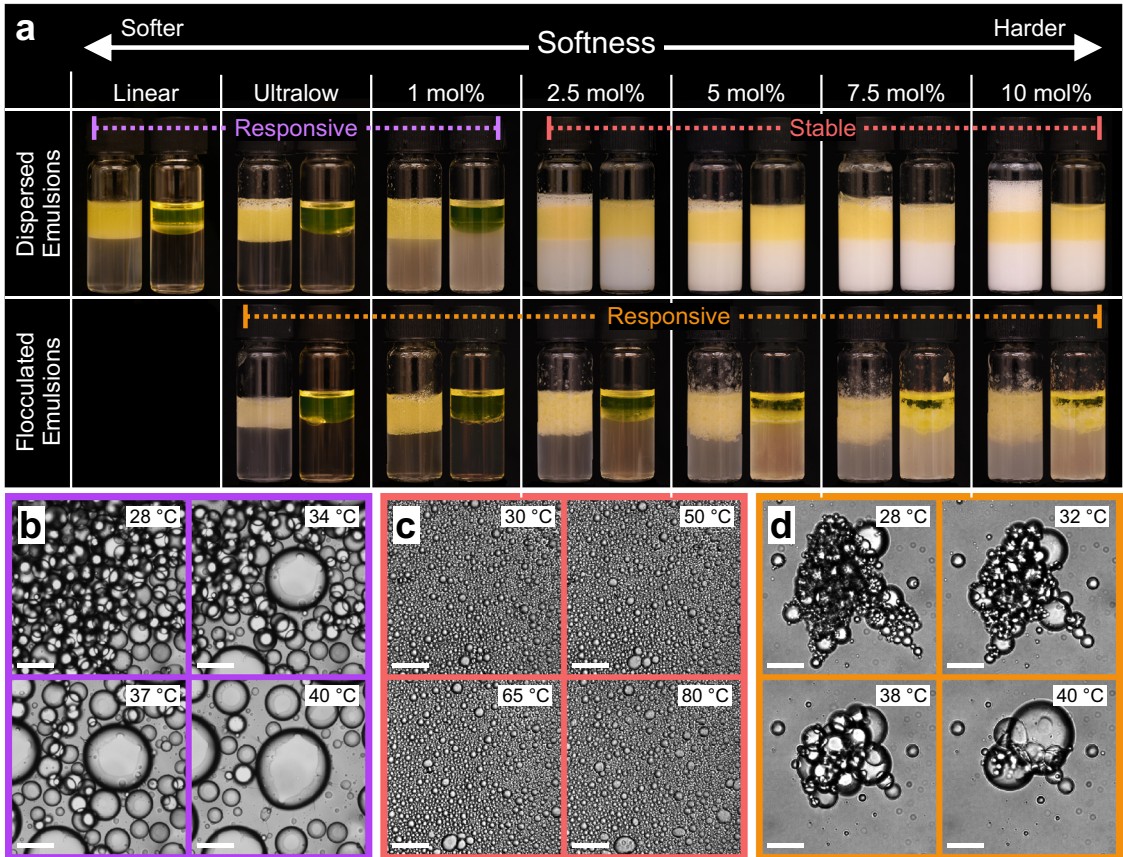

**Fig. 2 | Stimuli-responsive behaviour of dodecane in water emulsions stabilized by PNIPAM microgels. a** Vials of dispersed (top) and flocculated (bottom) emulsions stabilized by linear PNIPAM and PNIPAM microgels with increasing cross-linking densities prepared at 22 °C (left) and after storage at 55 °C for 4 h (right). The emulsions show creaming due to the density mismatch between dodecane and water. No flocculated emulsions were obtainable for linear PNIPAM. **b**–**d** Optical microscopy images as a function of temperature of dispersed emulsions, stabilized by (**b**) ultra-low crosslinked (ULC) microgels and (**c**) 5 mol% microgels, and (**d**)

flocculated emulsions, stabilized by 5 mol% microgels. We classify the emulsion behaviour into three regimes. Dispersed emulsions stabilized by low crosslinked microgels, or linear polymers are responsive and break above $T_{VPT}$ (purple frame). Dispersed emulsions stabilized by microgels with higher crosslinking densities remain stable and most of the droplets do not coalesce even up to 80 °C (red frame). All flocculated emulsions, on the other hand, destabilize above $T_{VPT}$ (orange frame). Scale bars: 50 μm.

$G^{s\prime\prime}$ from below $T_{VPT}$ to above $T_{VPT}$, Fig. 1g, and the interface does not fluidise, although $G^{s\prime}$ marginally decreases. The $\gamma_O$ sweep supports this, Fig. 1c (red symbols), as the interface retains yielding behaviour.

For regularly crosslinked microgels, 1 mol% to 10 mol% crosslinker, the respective interfaces become an order of magnitude more elastic (Fig. 1d–f). The elasticity for interfaces with 1 mol% crosslinked microgels (Fig. 1d (blue symbols)) is $G^{s\prime} = 1.42(2) \times 10^{-3}$ Pa m at $\gamma_O = 0.01$, and $G^{s\prime\prime}$ more noticeably overshoots. Upon increasing $T$, Fig. 1h, the interface remains elastic, with $G^{s\prime}$ even rising to $1.7 \times 10^{-3}$ Pa m. Solid-like behaviour is correspondingly observed in the high-$T$ strain amplitude sweep (Fig. 1d (red symbols)). Upon increasing crosslinking density, the interfacial rheological behaviour remains qualitatively the same (Fig. 1e, f) a phenomenology we further reproduce for different microgel concentrations (Supplementary Figure 3, detailed discussion in Supplementary Information). The interfacial elasticity increases, $G^{s\prime} = 1.75(1) \times 10^{-3}$ Pa m at 5 mol% and $G^{s\prime} = 2.68(2) \times 10^{-3}$ Pa m at 10 mol%, with a sharpening rise in $G^{s\prime\prime}$. As $T$ is raised, the increase in $G^{s\prime}$ is smaller (Fig. 1i, j); such that the high and low-$T$ strain amplitude sweeps become closer as crosslinking density increases. Noteworthily, the response of linear PNIPAM is below the resolution limit of our set-up and could thus not be measured (Supplementary Fig. 4).

From the ULC to 10 mol% crosslinked microgels, alongside an increase in elasticity, there is a change in the nature of yielding towards

a drop in $G^{s\prime}$ with a sharp rise in $G^{s\prime\prime}$ (cf. Fig. 1c, f). This suggests a more well-defined onset of irreversible plastic deforming system with increasing macroscopic strain[49], reminiscent of a jammed system with dynamic heterogeneity[50] compared to an entangled polymeric system[49]. This is consistent with the interfacial morphology of the microgels[51], which we will discuss later, suggesting that interfacial shear rheology is an effective probe of lateral microgel interactions. To summarize, upon increasing temperature the interfaces do not fluidise or significantly weaken, with regularly crosslinked microgels even becoming more elastic. This implies that the surface coverage and lateral interactions do not alter at $T_{VPT}$ due to, e.g., shrinkage, desorption, or aggregation (Fig. 1a). Therefore, the previous class of explanations are insufficient to capture the behaviour of thermo-responsive microgel-stabilised emulsions.

### Occurrence or absence of stimuli-responsive behaviour is linked to the microgel's softness

Our interfacial rheology measurements on flat interfaces probe changes in the lateral structures of microgel monolayers whereas changes in vertical structure, such as microgel shrinking along the vertical direction, escape experimental verification. In addition, emulsions may also be stabilized by multilayers or monolayers shared by two emulsion droplets. Thus, we now explore how the stabilizing microgel's architecture, softness, and interfacial morphologies affect the respective macroscopic emulsion behaviour.

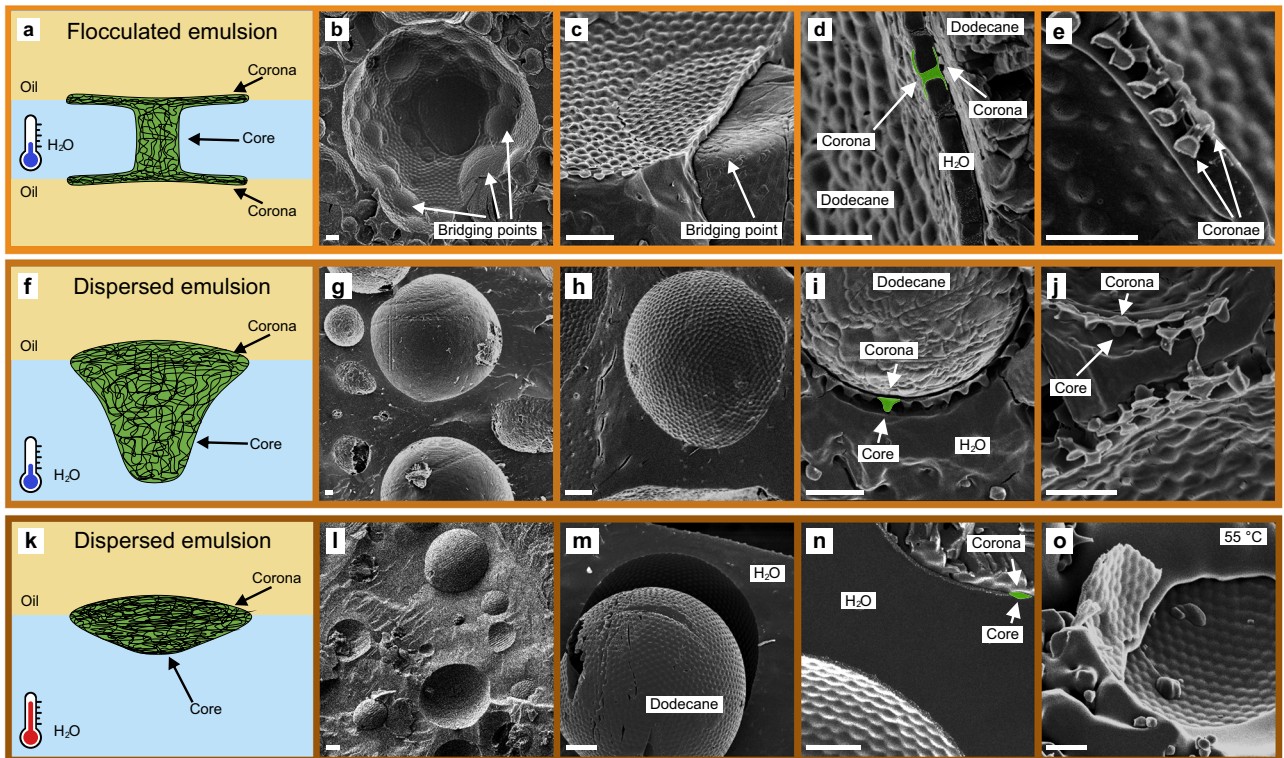

**Fig. 3 | Representative Cryo-SEM images of dodecane in water emulsions stabilized by PNIPAM microgels, from droplet level to particle level.**
**a**, **f**, **k** Schematic illustration of the stabilizing microgel morphologies for each emulsion type. **a**–**e** Flocculated emulsions are characterized by distinctive bridging points where the microgels are adsorbed to two oil droplets. These bridging microgels assume a characteristic morphology with two individual coronae formed at each liquid interface (**a**, **d**, **e**). Coalescence is prevented by a ~ 330 nm thick barrier consisting of microgels and water. **f**–**j** Dispersed emulsion droplets are characterized by a microgel monolayer and the droplets remain separated. The microgels adsorb only to one interface and assume a characteristic core-corona morphology with a swollen core extending into the water phase, inhibiting coalescence. **k**–**o** Dispersed emulsions droplets after storage at 55 °C for 4 h followed by immediate freezing using a liquid nitrogen slush. No significant change in lateral microgel assembly is observed compared to the samples stored at room temperature but the core part of the microgel exposed to the water side shrinks and the microgels transform to a flattened morphology. **d**, **i**, **n** Differences in characteristic microgel microstructures are highlighted with a green overlay. Scale bars: (**b**, **c**, **g**, **h**, **l**, **m**) 2 μm, (**d**, **e**, **i**, **j**, **n**, **o**) 1 μm.

We tune the softness of the stabilizing microgels by varying their crosslinking densities, which is known to systematically tune their swelling (Supplementary Fig. 1a, b)[52], elasticity[52–54] and compressibility at liquid interfaces (Supplementary Fig. 1c)[52,55]. We compare the full polymer-to-colloid range (Fig. 2a), starting from linear polymer via ULC microgels towards regular microgels with increasing crosslinking densities, which approach a colloid-like behaviour. Second, we investigate the effect of the microgel morphologies once adsorbed to the liquid interface by comparing two emulsion types, dispersed and flocculated emulsions. In dispersed emulsions, droplets can freely move within a continuous phase. Such emulsions are obtained using low-shear vortex mixing[40,56] with sufficient stabilising microgels (Fig. 2a, top). On the other hand, in flocculated emulsions, the droplets form aggregates without coalescence (Fig. 2a, bottom). They are obtained either using low-shear vortex mixing combined with a low concentration of stabilizing microgels or by high-shear emulsification using a rotor-stator setup, a frequently used emulsification method[27,56–58]. At room temperature, all types of emulsions stabilized by any of the investigated microgels are stable for months. For linear polymer, no flocculated emulsions are obtained and dispersed emulsions are not long-term stable at room temperature as they de-emulsify within hours (Supplementary Fig. 5)[32].

We then characterize the macroscopic response by comparing the vials containing each emulsion at room temperature (Fig. 2a, left) to the same emulsions stored at 55 °C for 4 h (Fig. 2a, right). We further probe the destabilisation dynamics by investigating the evolution of emulsions with increasing temperature on the droplet level using optical microscopy equipped with a temperature stage (Fig. 2b, d). We observe a striking difference between the stimuli-responsive behaviour, which we classify into three different types: First, dispersed emulsions stabilized by either linear polymer or low-crosslinked microgels destabilize and an oil phase (dyed yellow) is visible on the sample's top (Fig. 2a, purple frame), which has been reported previously[32]. We observe the coalescence of the oil droplets when heating above $T_{VPT}$ (Fig. 2b, Supplementary Movie 1). Second, dispersed emulsions stabilized by higher crosslinked microgels do, surprisingly, not show any macroscopic change after heat treatment (Fig. 2a, red frame). They remain mostly unaffected even by temperature increases up to 80 °C and only rare instances of coalescence are observed, despite droplets being in close contact (Fig. 2c, Supplementary Movie 2). These emulsions also remain stable for weeks stored at 55 °C (Supplementary Fig. 6) and under mild shaking (Supplementary Fig. 7, detailed discussion in Supplementary Information). Third, flocculated emulsions destabilize independently of the crosslinking density of the stabilizing microgels (Fig. 2a, orange frame), in agreement with previous reports[9,29–31,36,39–41]. We observe the coalescence of the oil droplets at the bridging points, where droplets are bonded when heating above $T_{VPT}$ (Fig. 2d, Supplementary Movie 3). In addition, the de-emulsification rate in flocculated emulsions is slower for higher crosslinked microgels (Supplementary Figure 8). To summarize, the stimuli-responsiveness is correlated with the softness of the stabilizing microgels as well as the emulsion type, dispersed or flocculated. Our results suggest that it is the microgel-mediated

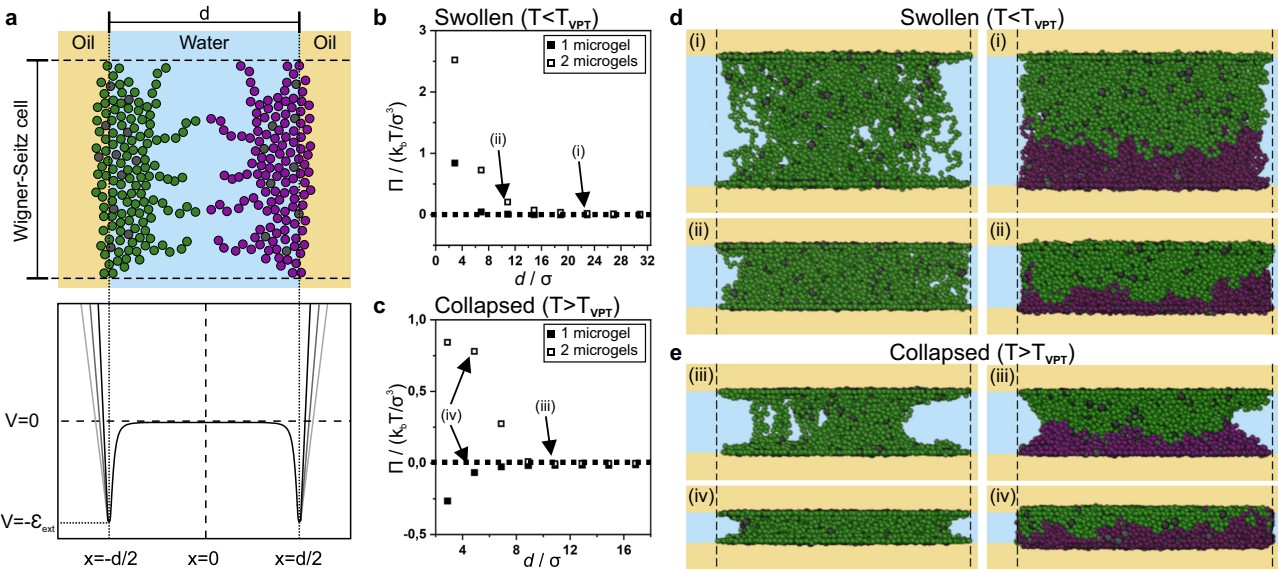

**Fig. 4 | Monomer-resolved Brownian dynamics simulations of regular thermo-responsive microgels confined between two planar liquid interfaces and a Wigner-Seitz cell. a** Schematic illustration of the simulation set-up (top) and the corresponding interfacial monomer potential (bottom). The monomers are shown as green/purple spheres and the crosslinker as grey spheres. The potential of the oil phase was varied (black, dark grey and grey curves) to allow more beads (size σ) in the oil phase, mimicking changes in microgel wettability. **b, c** Osmotic pressure $\Pi$ exerted by the microgel onto the liquid interface as a function of distance $d$ between the two liquid interfaces of one microgel adsorbed to both interfaces

(filled, representing flocculated emulsions) and two confined microgels each adsorbed to one interface (hollow, representing dispersed emulsions) in the swollen (**b**) and in the collapsed (**c**) state. A positive $\Pi$ corresponds to a repulsive force of the microgels onto the two liquid interfaces. **d, e** Representative snapshots of the microgels at different compression states in the swollen (**d**) and collapsed (**e**) states. The Roman numerals (i-iv) link the snapshot to the corresponding data points in (**b, c**). For one microgel, we notice the formation of a catenoid structure upon collapsing.

interaction between, rather than within, interfaces that is key to the destabilisation mechanism.

## Responsive emulsion behaviour is linked to the stabilizing microgel morphologies

In the following, we will first address how emulsions stabilized by the same microgels with higher crosslinking densities (here, 5 mol% crosslinker) can either be stable in the case of dispersed emulsions or responsive in the case of flocculated emulsions (Fig. 2c, d). To study the difference between both emulsions at the particle level, we turn to cryo-scanning electron microscopy (cryo-SEM). We filled copper rivets with the emulsions followed by rapid freezing in liquid nitrogen slush and breaking of the rivet under vacuum to reveal a cross-section through the emulsion[59]. The fracture typically occurred at the microgel/oil interface, revealing the assembly of the microgels at this interface (Fig. 3). Hereby, the microgels remain anchored in the water phase and the part exposed to the oil phase is revealed, whereas in the oil phase, the imprint of the microgels is visible (Fig. 3m). More rarely, the fracture occurred at both the microgel/oil and microgel/water interface. This fracture mode discloses the full 3D nature of the microgels at the liquid interface.

When comparing flocculated and dispersed emulsions, we observe significant differences; in particular, flocculated emulsions have characteristic bridging points connecting droplets (Fig. 3b, c), as described previously[57,60–62]. The cross-sectional images reveal that at these bridging points the microgels are adsorbed to both oil droplets and assume a pronounced corona at each droplet interface driven by a competition between internal elasticity and gain in the interfacial energy due to interfacial adsorption (Fig. 3a, d, e). In addition, they protrude less into the oil phases and appear more flattened (Fig. 3c). At room temperature, these microgels maintain a thin water layer (thickness: ~330 nm, Fig. 3d) between the oil droplets, which seemingly prevents coalescence.

Dispersed emulsions are characterized by a microgel monolayer and the absence of any bridging points (Fig. 3f–j). The adsorbed

microgels form a close-packed monolayer (Supplementary Fig. 9, detailed discussion on the microgel assembly in Supplementary Information) and assume a core-corona morphology at the oil/water interface while the bulk of the microgel is exposed to the water phase (Fig. 3i, j), in agreement with in-situ AFM measurements[63]. The microgel monolayer keeps the oil droplets separated (Fig. 3g, h) and prevents coalescence (Fig. 2c). We further image the same emulsion after storage at 55 °C for 4 h followed by immediate freezing using liquid nitrogen slush. We observe no qualitative change in the microgel assembly in the top view, no lateral shrinking or signs of any multilayer formation (Fig. 3k–o), corroborating our temperature-independent interfacial rheological data (Fig. 1e, i). The part exposed to the water phase, however, appears more flattened (cf. Fig. 3i, n), in agreement with previous predictions from simulations[43,45,46] and experiments[44,45,63].

## Above $T_{VPT}$, microgels adsorbed to two interfaces induce an attractive force between them, which leads to coalescence

To reveal how different microgel morphologies may lead to a loss of stability above $T_{VPT}$, we model the stability of emulsions stabilized by microgels adsorbed to either one or two interfaces using monomer-resolved Brownian dynamics simulations. The microgels are synthesized in-silico by assembling monomer and crosslinker units, a technique pioneered by Gnan et al.[64] in 2017 that has since been established as a numerical model microgel system[43,45,46,65–67]. The model makes it possible to tune swelling and collapsing of the microgel by changing the interactions between their monomer units[64]. Here, we equilibrate either one or two microgel(s) in the swollen state ($T < T_{VPT}$) between two attractive planar oil/water interfaces (Fig. 4a). Next, we bring the two interfaces together to mimic the approach of two emulsion droplets and we measure the osmotic pressure ($\Pi$) exhibited by the microgel onto each liquid interface. Upon approach, we measure an increasing positive osmotic pressure with decreasing separation distance ($d$) for both one and two microgels (Fig. 4b). A positive osmotic

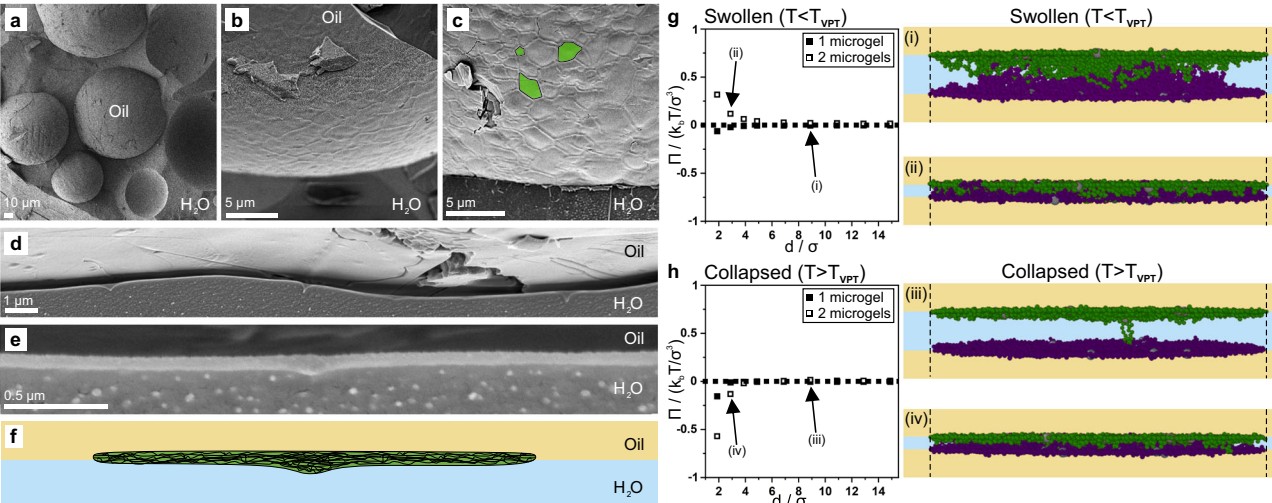

**Fig. 5 | Dodecane in water emulsions stabilized by ultra-low crosslinked (ULC) PNIPAM microgels. a–c** Representative cryo-SEM images of ULC microgel at the emulsion interface in top view revealing the pronounced spreading of ULC microgels at the liquid interface. Compared to regular microgels (Fig. 3f–j), no ordered lattice is visible for ULC microgels and the area they occupy at the interface is ill-defined in size and shape (**b, c**). The top-view morphology of individual microgels is highlighted with green overlays (**c**). **d–e** Microgel morphology in cross-section accompanied by a schematic illustration (**f**). **g, h** Monomer-resolved

Brownian dynamics simulations: Osmotic pressure *Π* exerted by either one (filled, representing flocculated emulsions) or two (hollow, representing dispersed emulsions) ULC microgels onto the liquid interface as a function of distance *d* in the swollen (**g**) and in the collapsed (**h**) state. A positive *Π* corresponds to a repulsive force of the microgels onto the two liquid interfaces. (i-iv) Representative snap-shots of two ULC microgels at different compression states in the swollen (i-ii) and collapsed (iii-iv) states.

pressure corresponds to a repulsive force between the emulsion droplets induced by the stabilizing microgels. Further, at $T < T_{VPT}$, the microgels remain swollen and appear to maximize the occupied volume (Fig. 3e). These results corroborate our experimental observations as both flocculated and dispersed emulsions do not coalesce at low temperatures (Fig. 2c, d).

We repeat the approach of the two interfaces, but with microgels in the collapsed state, mimicking the approach of two emulsion droplets above $T_{VPT}$ (Fig. 4c, e). For a single microgel adsorbed to two interfaces, we instead measure a negative *Π* upon approach, corresponding to an attractive force between the two emulsion droplets induced by the microgel (Fig. 4c). In contrast, two microgels confined between the two interfaces continue to give rise to a positive osmotic pressure upon compression, corresponding to a repulsive force between the emulsion droplets (Fig. 4c). We qualitatively reproduce this behaviour for different microgel wetting conditions (Supplementary Figure 10, detailed discussion in Supplementary Information), suggesting that any potential changes in microgel wetting may not be the driving force behind the temperature-induced destabilisation. We conclude that above $T_{VPT}$, the microgels located at bridging points between emulsions (Fig. 3a, e) pull the two emulsion droplets together, leading to a collapse of the thin water layer between the emulsion droplets (Fig. 3d), and inducing coalescence in flocculated emulsions (Fig. 2a, d, orange frame). On the other hand, the repulsive force measured for two microgels explains why dispersed emulsions, characterized by a microgel monolayer (Fig. 3f–j), are stable against coalescence, even at temperatures well above $T_{VPT}$ (Fig. 2a, c, red frame).

**Flattened microgel morphologies enable stimuli-responsive dispersed emulsions**

We will now address why dispersed emulsions stabilized by microgels with lower crosslinking densities display a stimuli-responsive behaviour (Fig. 2a, b) whereas their higher crosslinked counterparts are insensitive to temperature stimuli (Fig. 2a, c). We use cryo-SEM to shed light on the morphology and assembly of ULC microgels confined at the droplet interface (Fig. 5a–f). Visualizing individual ULC microgels remains a challenge as they are known to flatten at liquid interfaces

into pancake shapes[45,51,54]. In addition, they tend to intertwine with each other, which makes them appear more like a continuous polymer film instead of a particle monolayer[51]. We thus use an additional sub-limation step to disclose the particulate character of the ULC microgels in the top view (Fig. 5b, c). We observe that the shape of ULC microgels and the degree to which they expand at the liquid interface is ill-defined, highlighted with green overlays (Fig. 5c) and they do not assemble into an ordered lattice (Fig. 5b, c). This may be attributed to the size polydispersity and the sparse distribution of crosslinking points within ULC microgels, leading to inhomogeneous spreading[51]. In cross-section, the ULC monolayer appears similar to a continuous polymer film and only minor extensions into the aqueous phase reveal their particle nature (Fig. 5d–f).

Next, we repeat our monomer-resolved Brownian dynamics simulations for ULC microgels and we again measure the osmotic pressure (*Π*) exhibited by the microgel onto each liquid interface upon approach (Fig. 5g, h). We capture the more flattened morphology of ULC microgels by decreasing their crosslinking density as well as by increasing the Wigner-Seitz cell, allowing them to spread more at the liquid interface (Fig. 5i–iv, Supplementary Fig. 11). Next, we repeat our monomer-resolved Brownian dynamics simulations for ULC microgels and we again measure the osmotic pressure (*Π*) exhibited by the microgel onto each liquid interface upon approach (Fig. 5g, h). We capture the more flattened morphology of ULC microgels by decreasing their crosslinking density as well as by increasing the Wigner-Seitz cell, allowing them to spread more at the liquid interface (Fig. 5i–iv, Supplementary Fig. 11). We opted for a crosslinking density of 0.3%[45] due to computational constraints, striking a balance that enables a low crosslinker density while still retaining the essential features of a microgel. In the swollen state, we measure a positive osmotic pressure upon approach for 2 microgels, explaining the stability of dispersed emulsions at room temperature. Interestingly, we measure a mildly attractive force for one microgel, predicting that flocculated emulsions may not be long-term stable. In the collapsed state, above $T_{VPT}$, negative osmotic pressures are obtained for both one and two microgels, corroborating our experimentally observed macroscopic destabilisation for both emulsion types (Fig. 1a, b).

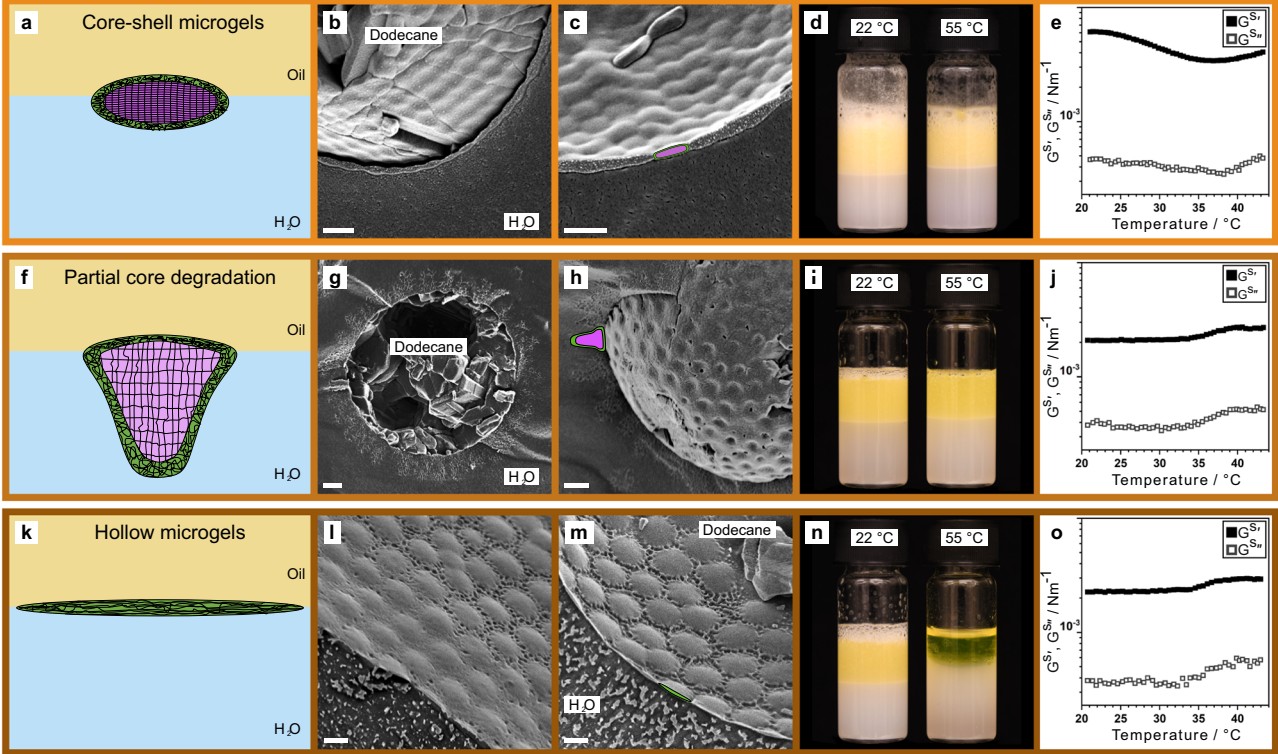

**Fig. 6 | Emulsion stability in relation to the microgel internal architecture and characteristic interfacial morphology.** Dodecane in water emulsions stabilized by core-shell microgels (**a**–**e**), whose inner microgel core (illustrated in purple) is either partially degraded by cleaving approximately 20% of the crosslinks (**f**–**j**) or fully degraded (k-o) to obtain hollow microgels. **a**–**c**, **f**–**h**, **k**–**m** Cryo-SEM images and corresponding schematic illustrations of the characteristic interfacial morphologies of core-shell microgels (**a**–**e**), partially degraded core-shell microgels (**f**–**j**) and hollow microgels (**k**–**o**). **c**, **h**, **m** Overlays illustrate the position of the microgel core (purple) and its shell (green). **d**, **l**, **n** Corresponding emulsion at room temperature and after storage at 55 °C for 4 h. **e**, **j**, **o** Linear viscoelastic response with increasing temperature at strain amplitude $\gamma_O = 0.01$ with storage ($G^{s\prime}$, filled) and loss ($G^{s\prime\prime}$, open) moduli. Scale bars: 500 nm.

## Characteristic interfacial microgel morphologies enable stimuli-responsive emulsions

Our combined cryo-SEM and molecular dynamics investigations reveal that the microgel morphologies and their interactions between interfaces are key to the destabilisation mechanism. In the previous sections, we tuned the microgel morphologies by systematically increasing their crosslinking densities. Here, we will utilize a second approach to tune the internal architecture and morphology of the stabilizing microgels by synthesizing core-shell microgels whose core can be chemically degraded[67,68]. This enables us to gradually change the internal architecture of the same microgels from hard core-shell microgels, via core-shell microgels with partially degraded cores, towards hollow microgels after fully degrading the cores to explicitly reveal the core's role in emulsion stability (Fig. 6, Supplementary Fig. 12).

The core-shell particles consist of a densely crosslinked PNIPAM core with 10 mol% degradable crosslinker and a smaller PNIPAM shell with 5 mol% non-degradable crosslinker (Fig. 6a). Cryo-SEM images at the emulsion interface reveal that, instead of a typical core-corona morphology, these core-shell microgels assume an oblate shape (Fig. 6a–c). Interestingly, this morphology matches analytical solutions of soft elastic spheres at liquid interfaces, hinting at a homogeneous distribution of the degradable crosslinker[69]. The corresponding emulsions show no sign of stimuli-responsiveness (Fig. 6d). From interfacial rheology, we notice two kinks in $G^{s\prime}$ with increasing temperature. The decrease in $G^{s\prime}$ at 25 °C can attributed to the $T_{VPT}$ of the inner microgel core (where the $T_{VPT}$ of the core is lower compared to regular microgels due to a different crosslinker), which may counteract the oblate deformation. The increase in $G^{s\prime}$ at 32 °C can be related to the $T_{VPT}$ of the microgel shell, potentially because of an increase of the microgel fraction absorbed to the liquid interface.

When partially degrading approximately 20 % of the crosslinker from the inner core[67], the microgels become less restricted and can adapt their shape. Like regular microgels (Fig. 3f–j), they assume a core-corona morphology at the liquid interface with a microgel core extending into the water phase. Further, we observe no macroscopic destabilisation after heat treatment (Fig. 6i) and interfacial rheology reveals a qualitatively similar behaviour compared to regular microgels (cf. Fig. 6h, Fig. 1i). Fully degrading the core leads to hollow microgels which spread along the liquid interface and deform into a flattened disk-like morphology (Fig. 6k–m). They cover 4 times more interfacial area compared to native and partially degraded core-shell microgels despite the drastically lower polymer content due to the core degradation. The corresponding emulsions are stimuli-responsive and destabilize above $T_{VPT}$ (Fig. 6n). We assume that their characteristic flattened morphology and the absence of a core extending into the aqueous phase cannot provide sufficient steric stabilisation above $T_{VPT}$, leading to coalescence like ULC microgels (Fig. 5). The similar response in interfacial shear rheology compared to microgels with partial core degradation further corroborates that it is not the interactions of the stabilizing microgels within, but instead the interactions between the interfaces, that determine the stimuli-responsive emulsion behaviour. To summarize, this core-shell microgel system allowed us to gradually change the internal architecture of the stabilizing microgels and confirm that the presence of microgel cores prevents stimuli-responsive de-emulsification of dispersed emulsions. Only after core degradation do the respective emulsions become stimuli-responsive.

## Discussion

In this article, we shed light on the destabilisation mechanism of stimuli-responsive emulsions stabilized by PNIPAM microgels.

Previously, the destabilisation mechanism has been linked to the characteristic volume phase transition of PNIPAM microgels. It was speculated that above $T_{VPT}$, the stabilizing microgels shrink laterally, which would lead to fluidised interfaces due to lower surface coverage or weaker interfaces due to potential microgel aggregation. However, our interfacial shear rheology reveals this fluidization ($G's<G's'$) does not occur at $T_{VPT}$ and that thermo-responsive destabilisation cannot be attributed to the assembly of an isolated interface, contradicting previous models. This establishes that the interaction between interfaces and the vertical, rather than lateral, microgel morphology is key.

We further find that stimuli-responsive emulsion behaviour is linked to the morphology of the stabilizing microgels and the type of emulsion, which we classify into three regimes. Flocculated emulsions are stimuli-responsive regardless of the internal architecture of the stabilizing microgels. Freeze-fracture cryo-SEM reveals that these emulsions are characterized by bridging points, where two droplets are joined. Here, the microgels are shared between both droplets and they form a corona at each interface. A thin water layer within the bridged region prevents coalescence at room temperature. However, Brownian molecular dynamics simulations reveal that once the temperature is increased above $T_{VPT}$, the microgels shared by two interfaces induce an attractive force, leading to coalescence. Previous work also reported a combination of mono- and bilayers within bridging points for softer microgels[57]. We infer that the bridging microgels still act as weak links, inducing coalescence. Notably, an absence of temperature-induced coalescence was reported for flocculated emulsions where the PNIPAM microgels contained additional charged comonomers[30,31,70]. Here the charged moieties, absent in our simulations, may counteract the attractive forces induced by a double-corona morphology microgel, which may explain why coalescence was only observed if the charges were neutralized[30,31,70].

In dispersed emulsions, the response depends on the architecture and interfacial morphology of the stabilizing microgels. Stimuli-responsive emulsions are obtained for low crosslinked or hollow microgels. Like linear polymers, they extend at the liquid interface and assume a flattened pancake morphology. While these emulsions are stable at room temperature, the flattened microgels collapse into a thin film above $T_{VPT}$, which is not sufficient to prevent coalescence. Brownian dynamics simulations show a negative osmotic pressure, i.e. attractive force, even for two microgels. By contrast, dispersed emulsions stabilized by more highly crosslinked colloid-like microgels do not coalesce upon temperature increase. We attribute the absence of stimuli-responsiveness to their characteristic core-corona morphology with a core extending into the water phase, creating a repulsive force between interfaces, even above $T_{VPT}$. To summarize, we find that the occurrence and absence of a thermo-responsive emulsion breaking mechanism are, therefore, linked to the morphology of the stabilizing microgels and not the interfacial properties of the assembly.

Upon coalescence, the interfacial area of the droplets is reduced leading to a lateral compression of the interface and additional effects may arise controlling the rate of coalescence. The shrinking area will eventually lead to failure of the microgel monolayer via either desorption, interfacial wrinkling, or the formation of multilayers. While these larger-scale structural effects are not probed by our Brownian dynamics simulations, post-coalescence imaging can indicate the possible mechanism in action. Soft microgels are known to be able to desorb from the liquid interface to the aqueous phase upon lateral compression[71]. This may cause the clear separation into an aqueous and oil phase after de-emulsification (Fig. 2a). On the other hand, interfacial wrinkling has been observed in Langmuir monolayers of regular microgels with higher crosslinking densities[72]. This may explain the formation of a microgel cluster, which after de-emulsification sits between the aqueous and oil phases (Fig. 2a, Supplementary Fig. 13). Confocal microscopy of said cluster reveals the microgel shells of the former emulsion droplets remain relatively intact even after the

encapsulated liquid is drained (Supplementary Fig. 13). Seemingly, the redispersion of the microgels into the aqueous phase is hindered. In addition, some oil droplets are found in the cluster, which likely became trapped during the de-emulsification process, leading to it remaining buoyant between the bulk oil and aqueous phases. To summarize, the tendency of more crosslinked microgels towards interfacial wrinkling instead of desorption may slow the de-emulsification of flocculated emulsions compared to softer microgels (Supplementary Fig. 8). In addition, monolayer failure through wrinkling may likely be an additional component that hinders coalescence, promoting the observed stable dispersed emulsions for more crosslinked microgels. This suggests that the microgel architecture and morphology affect coalescence dynamics in parallel to the effects initiating or preventing coalescence.

Our investigations highlight the importance of the nature of the stabilizing microgel particles, i.e. more polymeric vs more colloidal. The microgels' colloidal properties are fundamental to the long-term stability of Pickering emulsions. Emulsions stabilized by linear polymers are not long-term stable (Supplementary Fig. 5), which we attribute to the weak interface (Supplementary Fig. 4) as shown in recent work on dendronized polymers[73]. On the other hand, non-responsive emulsions are obtained if the stabilizing microgels are too hard with limited deformability. At the liquid interface, such microgels assume a core-corona morphology with a core extending into the aqueous phase that provides sufficient steric stability even above $T_{VPT}$. In our study, we find that ULC microgels, regular microgels with 1 mol% crosslinker and hollow microgels fulfil the balance between polymeric and colloidal properties required to enable stimuli-responsive emulsions. Their colloidal properties lead to an elastic interface providing long-term stability while their polymeric properties allow for them to spread and flatten at the liquid interface, enabling stimuli-responsiveness. On the other hand, microgels adsorbed to both droplets in flocculated emulsions enable stimuli-responsiveness regardless of their internal architecture. We believe that the importance of the microgel interfacial morphology and their polymer-colloid duality on the stimuli-responsive emulsion behaviour will also be of relevance to other stimuli-responsive systems. The presence of bridging microgels has been reported within thermo-responsive foams[53,74], which may similarly serve as weak links prompting the macroscopic foam destabilisation. Alternatively, emulsions[14] and foams[75] stabilized by pH-responsive microgels can be broken on demand upon change in pH. Similarly, we expect that in those systems the colloidal properties will ensure stability while the polymeric properties and softness of the microgels enable responsive behaviour.

This study also implies that the vertical structure of microgel stabilizers, not just the lateral structure, is of vital importance in stimuli-responsive interfacial behaviours. This aspect is often overlooked, which we believe is due to difficulties in experimentally assessing changes in vertical structure. Conventional interfacial techniques, such as interfacial rheology, pendant drop measurements or Langmuir methods typically probe changes in lateral structure whereas changes in vertical structure remain undisclosed. Thus, we see an opportunity to develop experimental techniques to directly measure changes in the vertical structure of interfacial monolayers, e.g., liquid phase atomic force microscopy combined with colloidal probes[76] or optical tweezers[77]. It is our hope that our article may trigger further research efforts in these directions.

## Methods
### Materials
N,N'-methylenebis(acrylamide) (BIS; 99%, Sigma Aldrich), ammonium persulfate (APS, Sigma Aldrich, 98%), potassium persulfate (KPS, Merck, >99%), N,N'-(1,2-dihydroxyethylene)bisacrylamide (DHEA, Merck, 97%), methacrylic acid (Merck, 99%), sodium periodate (99.8%), Trichloro(1H,1H,2H,2H-perfluorooctyl)silane (PFOCTS, 97%, Sigma

Aldrich), ethanol (Sigma Aldrich, >99.5%), linear poly(N-isopropylacrylamide) (PNIPAM, 10 kD, Sigma Aldrich) and Nile Red (>98%, Sigma Aldrich), hexane (≥99%, Sigma Aldrich) were used as received. N-Isopropylacrylamide (NIPAM; 97%, Sigma Aldrich) was purified by recrystallization from hexane (95%, Sigma Aldrich). Dodecane (99%, Acros organics) was passed through an alumina column twice. Water was double deionized using a Milli-Q system (18.2 MΩ·cm).

## Regular microgel synthesis
PNIPAM microgels were synthesized by surfactant-free precipitation polymerisation by reacting NIPAM with 5 mol% crosslinker BIS using the initiator APS[78]. In a 500 mL three-neck round bottom flask equipped with reflux condensers and stirrers, 2.83 g of NIPAM and the respective amount of BIS (1 mol%: 0.038 g, 2.5 mol%: 0.096 g, 5 mol%: 0.193 g, 10 mol%: 0.385 g) were dissolved in 249 mL of Milli-Q water. The solution was heated to 80 °C and purged with nitrogen gas. After 30 min equilibration, the nitrogen gas inlet was replaced by a nitrogen-filled balloon to sustain the nitrogen atmosphere. The reaction was initiated by adding 14.3 mg of APS dissolved in 1 mL of water. The microgels were cleaned three times by centrifugation and redispersion in water.

## Ultra-low crosslinked (ULC) microgel synthesis
ULC PNIPAM microgels were synthesized according to Virtanen et al.[79]. In a 250 mL two-necked round-bottomed flask, 0.071 mol/L NIPAM monomer was dissolved in 80 mL of distilled water and heated in an oil bath to 60 °C. A condenser was placed in the top inlet of the flask and nitrogen was bubbled in through the flask side arm for 20 min whilst it came to temperature equilibrium with the oil bath. In a separate vial, 0.0031 mol/L KPS was dissolved in 20 mL of water. After the 20 min equilibration time, the KPS solution was added to the flask to start the polymerization reaction and the reaction was left to proceed overnight. No crosslinking agent was added to the system and the ultra-low crosslinking seen in the final particles is attributed to hydrogen abstraction by the persulfate initiator as surmised in ref. 79.

## Core-shell to hollow microgels
Core-shell microgels, partial core degradation and hollow microgels were synthesized according to Vialetto et. al.[67].

## Core synthesis
5 g NIPAM monomer, 0.2237 g methacrylic acid co-monomer and 1.041 g of the cross-linker DHEA were dissolved in 400 mL of distilled water in a 1 litre three-necked, round-bottomed flask. The flask was fitted with a stirrer turning at 350 rpm, a water-cooled condenser and a nitrogen inlet and heated to 80 °C whilst continuously stirring and bubbling nitrogen through the system. 0.065 g KPS initiator was dissolved in 10 mL distilled water. After the 20 min equilibration, the KPS solution was added to the flask and the reaction was allowed to proceed for 4.5 h. The resulting particles were cleaned by centrifugation and subsequently redispersing them in distilled water: a process repeated ten times. After cleaning, the final dispersion had a mass fraction of 1.4 wt% and this was used as the sacrificial core for the hollow microgel particles.

## Core-shell microgel synthesis
To add a shell, 26.1 g of the dispersion (1.4 wt%) was placed in a 100 mL three-necked round-bottomed flask which itself was in an oil bath at 80 °C and nitrogen was bubbled through the system for 30 minutes. The other inlets of the flask contained a condenser and dropping funnel. In a separate vial 0.26 g NIPAM, 0.011 g methacrylic acid and 0.0197 g BIS were dissolved in 10 mL of water whilst in a second vial 0.003 g of KPS was dissolved in 1 mL of water. After the 30 minutes the contents of the NIPAM vial were placed in the dropping funnel and the initiator solution was added to the flask. The dropping funnel tap was

opened and the NIPAM solution slowly dripped into the flask. As there was no stirrer added to the flask, the nitrogen bubble flow was used to stir the reacting system. After 2 h the content of the dropping funnel had been added to the flask and the reaction was left to proceed for a further 2 h. The final particles were cleaned by centrifugation as described above.

## Core degradation
To degrade the core of these particles, sodium periodate (NaIO₄) was used to break down the DHEA cross-linker. For core-shell microgels with a partially degraded core, we added NaIO₄ in 10 times excess for 10 h, which according to Vialetto et al[67]. leads to a core degradation of ~20%. To obtain hollow microgels, we added NaIO₄ 400 times greater than that of the DHEA and the reaction was left to proceed for 48 h. The final particles were cleaned by 10 times centrifugation and redispersion to remove loose polymer chains.

## Microgel characterisation
The hydrodynamic diameter $D_H$ was measured by dynamic light scattering (Malvern Zetasizer Nano-ZS) and was 524 nm at 20 °C and 280 nm at 50 °C respectively (Supplementary Fig. 1a, d). We define the bulk swelling ratio $\beta = V_H (20\,°C) / V_H (50\,°C)$ (Supplementary Fig. 1b, e), where $V_H$ is the hydrodynamic volume. Further, the interfacial compressibility of microgel monolayers at air/water interfaces was measured using a Langmuir trough and the corresponding surface pressure was measured using a Wilhelmy plate (Supplementary Fig. 1c, f).

## Emulsion preparation
Emulsions were obtained by mixing 1 g aqueous microgel dispersions with 0.3 g dodecane (dyed with Nile Red for visualization) and emulsification using either vortex mixing[40,56,80] for 3 mins or a rotor stator[27,56–58] (IKA T10, S10N-5G) at 30,000 rpm for 3 mins. All emulsions were prepared at 22 °C. To obtain dispersed emulsions for more cross-linked microgels and avoid typically observed flocculation[32,57], the mass concentration of the microgel dispersion has been increased. Respective microgel type, concentration and emulsification methods for each experiment are shown in Supplementary Table 1. Emulsions with core-shell, partially degraded core-shell and hollow microgels were prepared at pH = 2.5 to protonate the acrylic acid groups[70].

## Optical characterization
The emulsions were sealed between hollow and flat glass slides, which had previously been functionalized with PFOCTS to reduce the interaction between the microgel-stabilized emulsions and the substrates. The emulsions were analysed using optical microscopy (Olympus BX50) equipped with a 20x objective and a temperature-controlled stage (Linkam LTS 350). The temperature of the stage was increased by 0.5 °C per minute up to 80 °C and images were taken each second. The temperature of the heating stage was additionally verified by heating a water-filled glass vial and measuring the temperature via a thermometer.

## Cryo-scanning electron microscopy
Cryo-SEM was carried out on a Zeiss Crossbeam 550 fitted with a Quorum Technologies PP3010T. Samples were prepared by filling glued copper rivets and freezing in nitrogen slush before mounting the rivet in a cryo stub under liquid nitrogen. Fracture of the sample was done by pushing the top of the rivet off in the vacuum of the Quorum PP3010T preparation chamber. The interfaces were sublimed for 5 min for regular microgels and 7 min for sublimation for ULC microgels at −90 °C, followed by sputtering a conductive Pt layer. Images were taken with the stage at −140 °C and with an accelerating voltage of 2 kV and a beam current of 200 pA using the secondary electron and Inlens detectors.

## Interfacial shear rheometry

Oscillatory interfacial shear rheology was carried out using a TA Instruments DHR-2 stress-controlled rheometer using a double-wall ring (DWR) geometry with a polyoxymethylene cup (inner radius 31 mm, outer radius 39.5 mm, depth 10 mm) and a platinum-iridium ring (diamond cross-section, 1 mm width, 35 mm radius). Surfaces were cleaned with ethanol and Milli-Q water. Milli-Q water was added as a lower phase. A PNIPAM microgel monolayer was created by spreading 10–100 μL of a PNIPAM suspension in a water-IPA mixture (9:1) at the air/water interface. The corresponding surface pressure per added microgel suspension volume has been measured on a Langmuir trough using a Wilhelmy plate. In Fig. 1, the concentration and amount added have been adjusted to reach a surface pressure of 24 mN/m for all interfaces. Interfaces stabilized with varying microgel concentrations – corresponding to different surface pressures – are discussed in the Supplementary Information and Supplementary Fig. 3. Then, the ring was lowered, ensuring that the interface was flat and pinned to the edges of the cup and ring. Filtered dodecane was then added as an upper phase, taking care not to disturb the interface. Temperature was controlled by a Peltier plate (set temperature from 20 °C to 55 °C), but to account for the thermal gradient across the geometry we report the temperature measured in the sub-phase via a thermocouple (RS Pro 1384 Temperature data logger). Interfaces were characterised at a set temperature of 20 °C using oscillatory strain sweeps before a temperature ramp and held to a set temperature of 55 °C while measuring the linear viscoelastic response. After temperature equilibration, the interface was again characterised using a strain sweep. Strain sweeps were performed in the low-frequency response plateau from a strain of 0.001 to 1 at 20 points per decade, logarithmically spaced, using one equilibration cycle and six measurement cycles per point. Measurements were primarily taken in a strain-controlled mode via a feedback loop after mapping of the bearing residual torque. For the ULC microgel, a closed-loop stress-controlled mode was used to improve the resolution limit[81] and a lower frequency to reduce the impact of instrument inertia; data was taken with torques selected to give an oscillatory amplitude sweep from a strain of 0.01 and a temperature sweep at a strain of 0.05.

## Brownian dynamics simulation

**In silico microgel synthesis.** Monomer-resolved Brownian dynamics simulations are performed to model the stability of flocculated and dispersed emulsions. Microgels are self-assembled from a binary mixture of bivalent monomer beads and tetravalent crosslinker beads, which was initially established by Gnan et al.[64] in 2017 and has since been frequently applied as a numerical microgel model system[43,46,65–67,82]. In brief, the in-silico synthesized microgels consist of a total number of 5500 monomer and crosslinker beads with implicit solvent. A monomer is covalently linked to either a monomer or to a crosslinker by springs, with a maximum of two bonds. Crosslinkers, on the other hand, have four such bonds. The crosslinking density is 4.5% for regular microgels and 0.3% for ULC microgels with regard to the total beads.

Although experimental evidence suggests an even lower cross-linking density for ULC microgels[51], below the 0.3% parameter used in our model, we made a considered compromise due to computational constraints. Our microgel model involves 5500 monomer and crosslinker units. Significantly reducing the crosslinking density, given our finite monomer count, could result in diverse polymeric structures that deviate from the characteristic microgel form. Additionally, a sparse distribution of crosslinkers poses the risk of creating an asymmetric microgel due to non-uniform crosslinker dispersion. Thus, we settled on a 0.3% crosslinker density, similar to the approach taken by Bochenek et al.[45]. This concentration strikes a balance, allowing for a low crosslinking density that reflects the emulsion behaviour observed in both ULC and 1 mol% microgels.

Importantly, it ensures that our in-silico model maintains the essential characteristics of a microgel.

In terms of all other interactions (next nearest neighbour interactions, etc.), monomers and crosslinkers do not differ and are therefore termed as beads in the following. The covalent bonds are described by a finite-extensible-nonlinear-elastic (FENE) potential with a characteristic energy scale $\varepsilon$, a maximal bond expansion $R_0 = 1.5\sigma$ and an effective spring constant $k_f = 15\varepsilon/\sigma^2$ [64,65,82]. The remaining bead–bead interactions are modelled by a repulsive Weeks–Chandler–Andersen (WCA) potential[83], which contains the size $\sigma$ of the repulsive monomers as a length and the repulsion strength $\varepsilon$, as the same energy scale as for the FENE potential. Hence, we choose $\sigma$ and $\varepsilon$ as units of length and energy respectively.

Further, an attractive bead-bead pair potential was added to model the thermo-responsivity of the microgels[64], which is given by Eq. (1)

$$V_\alpha(r) = \begin{cases} -\alpha\varepsilon & r \leq 2^{\frac{1}{6}}\sigma \\ \frac{1}{2}\alpha\varepsilon\left[\cos\left(\gamma\left(\frac{r}{a}\right)^2 + \beta\right) - 1\right] & 2^{\frac{1}{6}}\sigma < r < R_0\sigma \\ 0 & otherwise \end{cases} \quad (1)$$

where $\gamma = \pi\left(2.25 - 2^{\frac{1}{3}}\right)^{-1}$, $\beta = 2\pi - 2.25\gamma$ and $r$ is the distance between the bead centres. Importantly, the effective attraction strength is controlled by the parameter $\alpha$, which mimics the quality of the solvent in an implicit manner. $\alpha = 0$ describes good solvent conditions, as there is no attraction at all, reflecting the swollen state of the microgels below their volume phase transition temperature. On the other hand, $\alpha = 1$ describes strong attraction (relative to the bead repulsions) imitating poor solvent conditions and therefore mimicking the collapsed state of the microgel above their volume phase transition temperature (Supplementary Fig. 1g–i). For the connection between the effective attraction strength $\alpha$ and the temperature dependence on PNIPAM microgels, we refer to the work of Gnan et al.[64].

**Modelling of oil-water interfaces.** To mimic the effect of an oil–water interface, we add an external potential normal to the $x$-direction, Fig. 4a (lower). The external interface potential for each bead with a single water-oil interface is described by an effective Lennard-Jones (LJ) part and a steep linear part replacing the LJ divergence[46]. The former represents the water phase ($x > 0$) and the latter the oil phase ($x < 0$), with the single interface position at $x = 0$. For the Lennard-Jones potential,

$$V_{LJ}(x) = 4\varepsilon_{ext}\left[\left(\frac{\sigma_{ext}}{x}\right)^{12} - \left(\frac{\sigma_{ext}}{x}\right)^6\right] \quad (2)$$

an effective bead-interface interaction $\sigma_{ext}$ is introduced, with $\varepsilon_{ext}$ the attractive energy strength. At the matching point between the two parts, $x_a > 0$, the potential-value and the derivative (force) of both potential parts are chosen to be continuous, such that the external interface potential is given by

$$V_{ext}(x) = \begin{cases} V_{LJ}(x) & x > x_a \\ V_{LJ}(x_a) - (x_a - x)\frac{dV_{LJ}(x)}{dx}\Big|_{x=x_a} & x \leq x_a \end{cases} \quad (3)$$

The energetically favoured position for the beads is at the minimum of the interface potential at $x_{min} = 2^{1/6}\sigma_{ext}$. Physically, this corresponds to the effect of surface tension reduction by reducing the bare interface through bead adsorption. The assumed large difference in chemical potential between the oil and water phases is modelled by the steep increase in the potential for $x < x_a$. To ensure an increasing potential in the oil phase, $x_{min}$ is always larger than the matching point $x_a$. We have set $\varepsilon_{ext} = 5.5\varepsilon$, i.e. larger than the bead-bead interaction scale, to guarantee strong adsorption towards the interface[46]. Further,

$\sigma_{ext}$ is chosen as smaller than the bead size, $\sigma_{ext} = 0.5\sigma$, to form a relatively peaked interface, as assumed in the experiment. The matching point $x_a$ is slightly varied in our simulation and takes values between $1.110\sigma_{ext}$ and $1.115\sigma_{ext}$ to adjust the degree of softness of the interface potential, specifically decreasing repulsion from the oil phase. This leads to different fractions of the adsorbed beads in the two phases, with the two $\sigma_{ext}$ values leading to fewer or more beads in the oil phase respectively.

For two separated interfaces modelling approaching emulsion droplets, the total interfacial potential is given by the superposition of the two individual potentials at the corresponding shifted positions as

$$V_{interface}(x) = V_{ext}\left(x + x_{min} + \tilde{d}\right) + V_{ext}\left(-\left(x - x_{min} - \tilde{d}\right)\right) \quad (4)$$

with $\tilde{d} = d/2$ as half of the distance between the minima of each interface. The $x$ co-ordinate system of the effective interaction potential is now relative to the midpoint, between the two interfaces, as shown in Fig. 4a (lower).

**Interfacial confinement.** Experimental cryo-SEM images reveal that the microgels stabilizing the oil droplets are in a dense monolayer with dominant hexagonal symmetry. In the model, we introduce the confinement by a cylindrical Wigner-Seitz cell normal to the two interfaces. Its dimensions are chosen to qualitatively mimic the experimental data. We employ a radial softened force-shifted Lennard-Jones potential[83]. The bead confinement in the Wigner-Seitz cell is given by the external potential:

$$V_c(r) = \begin{cases} 4\varepsilon_c\left[\left(\frac{R_g}{R_c-r}\right)^{12} - \left(\frac{R_g}{R_c-r}\right)^6\right] + \Delta V_c(r) \ if & R_c - r \le R_g \\ \qquad\qquad 0, & otherwise \end{cases} \quad (5)$$

with

$$\Delta V_c(r) = -\left(R_c - r - R_g\right)\frac{\partial V_c^{LJ}}{\partial x}\left(X = R_g\right) \quad (6)$$

and

$$V_c^{LJ}(X) = 4\varepsilon_c\left[\left(\frac{R_g}{X}\right)^{12} - \left(\frac{R_g}{X}\right)^6\right] \quad (7)$$

Here $r$ is the radial distance between the cylinder centre and a bead and $X$ represents the distance between a bead and the wall of the Wigner-Seitz cell. The strength of confinement is $\varepsilon_c = 5\varepsilon$. $R_g = 20\sigma$ is the bulk radius of gyration[46] and $R_c$ is the Wigner-Seitz cell radius. For microgels with a crosslinking density of 4.5%, we have chosen $R_c = 40\sigma$ and for ULC microgels with 0.3% crosslinker, we chose $R_c = 60\sigma$.

**Simulation details and protocol.** The bead motion is simulated by Brownian dynamics, implying an implicit solvent. The short-time self-diffusion coefficient $D_O$ defines the Brownian time scale $\tau_B = D_O/\sigma^2$, which describes the time unit in our simulation. Using a finite time step of $\Delta t = 0.00009\tau_B$, the equations of motion are integrated by an Euler forward scheme. All of the Brownian dynamics simulations are performed with the HOOMD-Blue package[84] and are visualized by OVITO.[12]

We chose the described simulation protocol to qualitatively mimic the experimental emulsification process and the approach of two oil droplets. First, one or two microgels are equilibrated for 1000 $\tau_B$ in the fully swollen state ($\alpha = 0$) within the 3D Wigner-Seitz cylinder but in the absence of the two interfaces[46]. In the case of two microgels, the distance along the cylinder axis is large enough to avoid interactions.

In the next step, the external potential $V_{interface}(x)$ is turned on at a large separation distance of the liquid interfaces, $d = 4R_g$ for two microgels and $d = 2R_g$ for one microgel, initially avoiding any interfacial contact with monomers. Then the interface separation distance is gradually decreased in small steps of $\Delta d = 4\sigma$. This allows the microgels to naturally adsorb to the liquid interfaces and mimics the emulsification process. If two microgels are placed within the Wigner-Seitz cell, each adsorbs to one liquid interface only while with only one microgel within the cell it adsorbs to both interfaces (Fig. 4). For each new distance $d$, the system is equilibrated again for a time 300 $\tau_B$ and the measurements are run over a time window of 600 $\tau_B$.

We then repeat the measurement for microgels in their collapsed state (above their $T_{VPT}$) with $\alpha = 1$. We start from a swollen configuration ($\alpha = 0$) at the point the microgels adsorb to the liquid interfaces. $\alpha$ is then increased in steps of $\Delta\alpha = 1/6$. After each increase in $\alpha$, the system is equilibrated for 450 $\tau_B$. This simulation approach mimics experimental work where the emulsions are formed with the microgels in their swollen state followed by a gradual temperature increase (Fig. 2b,c,d). After equilibration at $\alpha = 1$, the distance $d$ is similarly decreased by an increment of $\Delta d = 2\sigma$ with the same equilibration and measurement times as for $\alpha = 0$.

Lastly, we repeat both approaches for different interfacial potentials. We slightly increase $x_a$ to obtain a shallower increase of the potential describing the oil phase, which leads to a higher percentage of beads located in the oil phase compared to the water phase mimicking a change in microgel wettability.

**Calculation of the osmotic pressure.** For each separation distance $d$, we measure the osmotic pressure $\Pi$ exerted by the microgels on each interface, which is the mean force between all beads divided by the area of the Wigner-Seitz cell.

$$\Pi = \frac{1}{\pi R_c^2}\left(-\sum_{i=1}^{N}\left\langle\frac{dV_{ext}(x)}{dx}\bigg|_{x=x_i-d-x_{min}}\right\rangle\right) \quad (8)$$

where $\langle\ldots\rangle$ denotes a time average. The osmotic disjoining pressures $\Pi$ acting on the interfaces as a function of the interfacial distance $d$ are shown in Fig. 4b,c and Fig. 5g,h. Dispersion forces between the two oil phases are negligible. They contribute to the osmotic pressure as $\Pi = -A / (6 \pi d^3)$ where $A$ is the Hamaker constant and $d$ the distance between two flat oil interfaces. In the units chosen in Fig. 4, the contribution is less than 1 percent even at the smallest distance $d$ if a typical value of $A = 10^{-20}$ J is taken for the Hamaker constant.

## Data availability
The data generated in this study are provided in the Supplementary Information/Source Data file. Data is available from the authors upon request. Source data are provided with this paper.

## Code availability
The code is available at: https://github.com/ishamalhotra612/Stimuli-Responsive_Emulsions.

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

## Acknowledgements

The authors acknowledge the following funding sources. Swiss National Science Foundation (Project-ID P2SKP2_194953), M.R.; Marie Sklodowska-Curie Individual Fellowship (Grant No.101064381), M.R.; European Soft Matter Infrastructure (EUSMI-h2020, grant agreement no. 731019), M.R. and J.V; Forschungsgemeinschaft (DFG) under grant numbers LO 418/22-1, H.L. and J.K.; Humboldt foundation, I.M. The authors further acknowledge access to the Cryo FIB/SEM bought with the EPSRC grant EP/P030564/1. The authors thank Liesbeth Janssen, Job Thijssen, Johannes Menath, Nicolas Vogel and Giovanni Volpe for feedback and fruitful discussions. For the purpose of open access, the

## Author contributions

M.R. synthesized and characterized the microgel particles. M.R. and A.B.S. characterized the macroscopic and microscopic response of thermos-responsive emulsions using optical microscopy. M.R., J.A.R. D.R. and J.V. probed the interfacial rheological properties. M.R., T.G, F.H.J.L. and N.Y.D.L. investigated the microgel microstructure using cryo-SEM. J.K., I.M. and H.L. performed the Brownian dynamic simulations. M.R., S.F. and P.S.C. designed the experiments and supervised the study. All authors contributed to the writing of the manuscript. M.R. and J.K. contributed equally.

## Funding

## Competing interests

The authors declare no competing interests.
