## [Peer Review File · Nature Communications]

Interactions between interfaces dictate stimuli-responsive emulsion behaviourREVIEWER COMMENTS

Reviewer #1 (Remarks to the Author):

The authors use interfacial shear rheology to characterize mechanical properties of interfaces with adsorbed microgels. Their results indicate that the underlying reason for the temperature-induced destabilization, unlike reported previously, cannot be attributed to the assembly at the isolated interfaces. The authors propose that the interaction between interfaces and the vertical microgel morphology (and not its lateral morphology) plays a key role in the observed phenomena.

I suggest that the authors specify the applicability and limitations of this study. A particular gel parameters (a single set of parameters) is used in this study. It is well known that a particular gel morphology upon its adsorption at the interface depends strongly on its elastic properties; in some instances characteristic microgel "softness" is defined. What is the parameter range at which the results reported in this manuscript are expected to hold? What are the elastic properties of the microgel used as a function of temperature? It had been shown previously that G' varies significantly with temperature for the thermo-responsive hydrogels. It would be instructive to characterize elastic properties of a hydrogel particles used in this study in addition to the visco-elastic properties of the interfaces.

Further, to validate parameters choice in the computational modeling of microgels, I would expect reproducing volume phase transition in microgels with temperature change. In other words, reproducing experimental plot provided in Figure S1, at least with a few simulation points.

Minor comment: double-corona could be more clearly defined somewhat earlier in the text.

Reviewer #2 (Remarks to the Author):

The manuscript by Rey et al reports a very interesting study on the stability of emulsions stabilized by temperature-sensitive PNIPAM microgels. They investigate properties of interfaces and emulsions using different experimental and computational techniques. They argue that the presence of microgels that are attached to the water-oil interface of two different droplets is required to have a temperature dependent (in)-stability of emulsions.

I fully agree with the authors that the stimulus-dependent stability of emulsions and thus the coalescence of microgel-covered emulsion droplets is still an open question. I also agree that understanding the (in)stability of such emulsions is important for exploiting the stimulus-sensitivity of such emulsions. Thus, I think the ms will be suitable for publication in this journal.

However, I have several comments which - in my view - suggest / require major revisions.

The authors use the term "double-corona particle morphology". I find this term very misleading: at least to me, when reading the title and the text, I interpreted this term such that it indicated that the microgels have intrinsically a special "double-corona particle morphology" (i.e. even in bulk). This is, however, not the case. The authors use normal PNIPAM microgels, which have in solution a core-corona type morphology and which also leads to a core-corona (or fried-egg) like structure when adsorbed at an oil-water interface. What - I think - the authors want to express is that it is important that a microgel is adsorbed to the surface of two different droplets. It is this bridging, that is relevant. Of course, soft microgels will then have the core-corona (or fried-egg) like structure at both interfaces, but I would not call this "double-corona".

I suggest the authors skip the term "double-corona particle morphology" and rather speak of bridging microgels. I have the impression that the size of the bridging microgel orthogonal to the interface, which changes with temperature is relevant for the onset of coalescence.

I do agree with the authors' statement that experiments at flat monolayers showed that microgels mainly shrink orthogonally and do not leave the interface. However, when emulsion droplets coalesce and finally macroscopic phase separation occurs, microgels have to leave the interface. It has also been observed that microgels leave the interface when the monolayer is compressed in a Langmuir trough.

The authors claim, that emulsions where the droplets are stabilized by a (mono)layer of microgels and the where the droplets are separated, do not break at high temperature. Well I accept that the authors made this observation, but I have two concerns:

(i) I feel that this is a rather artificial state. Real emulsions will be subject to "mechanical" disturbance, be it shaking, flow etc. I would be very surprised if these emulsions will be stable under such conditions.

(ii) This observation is different from what has been reported in the literature and I do not think that the authors' observation can be generalized.

In our recent study (Petrunin, A. V.; Bochenek, S.; Richtering, W.; Scotti, A. Harnessing the Polymer-Particle Duality of Ultra-Soft Nanogels to Stabilise Smart Emulsions. *Phys Chem Chem Phys* 2022. <https://doi.org/10.1039/d2cp02700c>.) we found a behavior that is different from the observations by Rey et al. With ultra-low crosslinked (ULC) microgels one can prepare well dispersed emulsions (no flocculated droplets) but the emulsions break at high temperature. Also, emulsions prepared with microgels containing 1% BIS, non-flocculated droplets can coalesce.

In one of our earlier studies, we found that emulsions can break under conditions where the microgels are still colloiddally stable in the aqueous phase or when the microgels flocculate in the aqueous phase (Wiese, S.; Tsvetkova, Y.; Daleiden, N. J. E.; Spiess, A. C.; Richtering, W. Microgel Stabilized Emulsions: Breaking on Demand. *Colloids and Surfaces A* 2016, 495, 193–199.)

In this context, I also find the term "quasi-neutral PNIPAM microgel" misleading. I fully agree, that the microgels do not carry many charges, but the small number of charges make them colloidally stable at $T > VPTT$ (as they are prepared in surfactant-free precipitation polymerisation). I suggest that the authors do some further control experiments in the presence of salt at a salt concentration where the microgels flocculate when heated above the VPTT.

Furthermore, I suggest that the authors also include a discussion of their results in comparison to: Destribats, M.; Eyharts, M.; Lapeyre, V.; Sellier, E.; Varga, I.; Ravaine, V.; Schmitt, V. Impact of pNIPAM Microgel Size on Its Ability To Stabilize Pickering Emulsions. *Langmuir* 2014, 30 (7), 1768–1777. I think their observation of size effects fits nicely to the results obtained by Rey et al.

I also suggest discussing the images in Figure 2 to the images reported by Destribats et al. *Langmuir* 28, 37744 (2012).

Can the authors evaluate the cry-SEM images in more detail and determine e.g. the surface coverage for different cases?

I agree with the authors' statement "that the vertical structure of microgel stabilizers, ..., is of vital importance". This agrees with our recent report by Petrunin et al in *PCCP* 2022.) where we compare normal PNIPAM microgels, ultra-low crosslinked (ULC) microgels and linear polymer.

However, while I see the relevance of vertical structure and the presence of bridging, I have the impression that the situation is still more complex. I wonder if the authors would agree to this view on stimuli-sensitive microgel stabilized emulsions:

In order to have a stable emulsion one needs: separated droplets, covered by a dense monolayer of microgels and a vertical (orthogonal) size of the microgel that provides sufficient steric stabilization. That suggests a minimum size of the microgels. Keeping in mind that big microgels are slow, it will be difficult to avoid having some microgels adsorbed to two droplets (bridging) during formation of the emulsion.

The coalescence of droplets requires close contact of the oil-water interfaces. Here bridging microgels will play an important role as their vertical (orthogonal) size (which depends on temperature as well as the density of polymer segments will set the relevant length scale for separating the oil-water interfaces.

Finally, mobility of the microgels at the interface, the viscoelasticity of the microgel covered interface and the rate of microgel desorption upon lateral compression of the interface are important for the transition from dispersed to aggregated (bridged) droplets and the rate of coalescence of bridged droplets.

As further comments:

I think it makes sense to compare the bridging behavior of microgels attached to two oil surfaces with the properties of foams. There are reports e.g. by the Regine von Klitzing group on foams stabilized by PNIPAM microgels (e.g. *Soft Matter* 2022 <https://pubs.rsc.org/en/content/articlehtml/2022/sm/d2sm01021f> .) (I think there are more papers on that topic in the literature)

In thin films, the microgel can also be attached to 2 interfaces and bridges these interfaces.

Concerning the simulations, am I right that the simulations did not include dispersion forces between the oil phases? I do not expect that this changes the conclusions, but I wonder why this has not been taken into account?

Concerning Figure 3 c/e: I guess, I understand how the authors calculate the osmotic pressure, but I have a (may be naive) question. Once the microgel is completely collapsed, I expect that the excluded volume of the polymer segments leads to a repulsion between the interfaces, and coalescence is not possible as the microgel cannot be compressed further. In order to have coalescence, the microgel must be pushed out. Am I right?

As a final (very minor) note concerning liquid phase colloidal probe AFM experiments suggested by the authors in the last paragraph. We (or rather the Butt group) tried that (but with a hard colloidal probe, not covered by microgels) (Bochenek, S.; McNamee, C. E.; Kappl, M.; Butt, H.-J.; Richtering, W. Interactions between a Responsive Microgel Monolayer and a Rigid Colloid: From Soft to Hard Interfaces. *Phys Chem Chem Phys* 2021, 23, 16754–16766.) These are tricky experiments and challenging to be interpreted...

Reviewer #3 (Remarks to the Author):

The article of Rey and coworkers addresses the stability of emulsions stabilized by microgel particles. They claim to have unraveled the mechanism that determines the stability of these exciting formulations. Namely, they associate the increased stability with a double Corona particle morphology that enables stimuli-responsive emulsions.

This article addresses an interesting and timely question that appears to be unanswered today.

Unfortunately, the paper also has several shortcomings that make it unsuitable for publication in *Nature Communications* at the current stage. First and most importantly, the authors claim that the double-corona particle morphology (of microgels) is a crucial aspect leading to stable emulsions. The authors, however, need to explain what they mean by a double Corona particle morphology (?), ideally adding a sketch. The paper is vague about these questions, and the figures reveal little. For example, in Figure 2,

many panels from cryo-SEM are added, but only in two of those some color patches are added. Again, I need clarification on what they show or proof. Also, here, a sketch of the drawing of microgel-morphology would be helpful.

As a reviewer, I would appreciate a better-quality copy of the manuscript. For a review, the font is small and inconvenient to read. One should certainly improve the figures. The fonts e.g., in Figure 1 and plots, are tiny. The images of the vials in Fig 1 c were taken at an oblique angle, and the phase separation is hard to see. Figure 2 looks like a random collection of SEM images where, as mentioned before, only three small parts are highlighted with red color. The connection between the simulations shown in Figure 3 and the (yet-to-be-defined) double-corona morphology is difficult to grasp.

In summary, this paper could be suitable for publication if the main claim can convincingly be supported by experimental evidence. The latter will require the authors to revise the manuscript and make it concise and more understandable to work out the main point.

KEY REVISIONS

We first address the two key points, which have been brought forward by all three referees and editor.

Applicability to other systems:

We thank all three Referees for their helpful suggestions and encouragement to validate our findings for other microgel systems. We took these suggestions to heart and extended our study with two additional sets of microgels (variation in crosslinking density, core-shell towards hollow microgels) to cover the wide range of tuneable physical-chemical parameters in soft microgel systems. This allows us to obtain an in-depth resolution of the link between the architecture and morphology of the stabilizing microgels and their corresponding emulsion behaviour.

In the first set of microgels, **we vary their softness by changing their crosslinker concentration.** Adjusting the crosslinking density is well-known to tune their swelling ratio, elasticity, and compressibility at liquid interfaces. In the revised manuscript (including Fig. 1-5), we now compare the full polymer-to-colloid range, starting from linear polymer via ULC (ultra-low crosslinked) microgels towards regular microgels with increasing crosslinking densities from 1-10 mol%.

We first establish that the interfacial response of microgel monolayers does not significantly alter with increasing the temperature above T_{VPT} regardless of their crosslinking densities. All interfaces remain highly elastic and no fluidization ($G^s < G^{s''}$) occurs. Therefore, a thermo-responsive destabilisation cannot be attributed to the assembly of an isolated interface, contradicting previous models. On the other hand, interfaces stabilized by linear polymers were much weaker (below the resolution of our setup) and their corresponding emulsions are not long-term stable. This indicates that the colloidal character of the stabilizing microgels is key to ensure long-term stable emulsions.

We then explore how the architecture and morphology of the stabilizing microgels governs the behaviour of dispersed and flocculated emulsions. We find that all flocculated emulsions coalesce above T_{VPT} independent of the crosslinking density. For dispersed emulsions, however, by changing the crosslinking density of the stabilizing microgels we find that stimuli-responsive behaviour occurs only for softer microgels (1 mol% crosslinker and lower), while the ones stabilized with harder microgels (2.5 mol% crosslinker and higher) remain stable above T_{VPT} . Cryo-SEM investigations and Brownian dynamics simulations reveal that the difference between responsive and stable dispersed emulsions is related to the morphology of the stabilizing microgels. Harder microgels assume a characteristic core-corona morphology once adsorbed to the liquid interface, with a core extending into the aqueous phase that provides sufficient steric stabilization to prevent coalescence even above T_{VPT} . On the other hand, softer microgels (like ULC microgels) drastically expand at the liquid interface into a flattened pancake-like shape and they appear more like a polymer film. The strongly elongated and flattened morphology of those microgels enables them to collapse into a thin film above T_{VPT} , which does not provide sufficient steric repulsion to prevent coalescence. We conclude that the absence of a microgel core extending into the aqueous phase and instead a flattened, pancake-like shape is required to produce stimuli-responsive dispersed emulsions.

To underline this conclusion, we investigate a second set of core-shell microgels who's interior core can be chemically degraded to obtain hollow microgels. This enables us to **gradually change the internal architecture of the same microgels from hard core-shell microgels, via core-shell microgels with partially degraded cores, towards hollow microgels** after fully degrading the cores to explicitly reveal the core's role in emulsion stability. Core-shell microgels and partially degraded core-shell microgels have a microgel core extending into the aqueous phase. For this

microgel morphology, we do not find a stimuli-responsive emulsion behaviour as the cores provide sufficient steric stability. Once the core is fully degraded, the hollow microgels assume a flattened and extended morphology. The corresponding emulsions are stimuli responsive.

To summarize, extending our study to two additional sets of microgels enabled us to correlate the macroscopically observed emulsion behaviour to the architecture and morphology of the stabilizing microgels. In addition, our investigations highlight the importance of the stabilizing microgel's polymeric and colloidal properties to enable stimuli-responsive dispersed emulsions. The microgels' colloidal properties are key to obtain an elastic interface, inducing long-term emulsion stability. On the other hand, non-responsive emulsions are obtained if the stabilizing microgels are too hard with a limited deformability. At the liquid interface, such microgels assume a core-corona morphology with a core extending into the aqueous phase that provides sufficient steric stability even above T_{VPT} . In our study, we find that ULC microgels, regular microgels with 1 mol% crosslinker and hollow microgels fulfil the balance between polymeric and colloidal properties required to enable stimuli-responsive emulsions. Their colloidal properties lead to an elastic interface providing long-term stability while their polymeric properties allow for them to spread and flatten at the liquid interface, enabling stimuli-responsiveness. We believe that the importance of the microgel interfacial morphology and their polymer-colloid duality on the stimuli-responsive emulsion behaviour will also be of relevance to other stimuli-responsive systems such as pH-responsive emulsion and foams.

In the light of the newly acquired data, we restructured the manuscript. Here, we briefly summarize the key changes.

- We split former Figure 1 into two Figures. The new Figure 1 now compares the interfacial response of microgels with varying crosslinking densities, showing the absence of any fluidization ($G^S' < G^S''$) above T_{VPT} , challenging previous models.
- Figure 2 compares the behaviour of flocculated and dispersed emulsions stabilized with microgels with different softness. We find three regimes: Flocculated emulsions are stimuli-responsive regardless of crosslinking density while dispersed emulsions stabilized by softer microgels are responsive while the ones stabilized by harder microgels remain stable.
- In Figure 5, we investigate the morphology of ULC microgels and use Brownian dynamics simulations to explain why their interfacial morphology enables stimuli-responsiveness.
- In Figure 6, we investigate the morphology and corresponding emulsion stability of core-shell microgels, whose core can be gradually degraded towards hollow microgels. This microgel system clearly demonstrates that the microgel core extending into the aqueous phase provides sufficient steric stabilisation even above T_{VPT} , preventing coalescence.
- We extended the Discussion section, where we discuss the importance of the polymer-colloid duality of microgels to ensure long-term stable, yet stimuli-responsive emulsions.

Double-corona morphology:

We apologize for the non-intuitive term “double-corona morphology”. As all 3 referees point out, “double-corona morphology” could be misunderstood that these microgels also have two coronae in bulk. What we instead meant with the term was to describe the morphology microgels assume when they simultaneously adsorb to two droplets, forming an extended polymer corona on each liquid interface. To avoid confusion, we have decided to not use the term “double-corona

morphology” anymore in the manuscript, and instead refer to those microgels as “bridging microgels”. In addition, we changed the title of the manuscript to:

“Characteristic interfacial microgel morphologies enable stimuli-responsive emulsions“

REVIEWER COMMENTS

Reviewer #1 (Remarks to the Author):

The authors use interfacial shear rheology to characterize mechanical properties of interfaces with adsorbed microgels. Their results indicate that the underlying reason for the temperature-induced destabilization, unlike reported previously, cannot be attributed to the assembly at the isolated interfaces. The authors propose that the interaction between interfaces and the vertical microgel morphology (and not its lateral morphology) plays a key role in the observed phenomena.

I suggest that the authors specify the applicability and limitations of this study. A particular gel parameters (a single set of parameters) is used in this study. It is well known that a particular gel morphology upon its adsorption at the interface depends strongly on its elastic properties; in some instances characteristic microgel “softness” is defined. What is the parameter range at which the results reported in this manuscript are expected to hold? What are the elastic properties of the microgel used as a function of temperature? It had been shown previously that G' varies significantly with temperature for the thermo-responsive hydrogels. It would be instructive to characterize elastic properties of a hydrogel particles used in this study in addition to the visco-elastic properties of the interfaces.

We thank Referee 1 for their helpful suggestions and encouragement to validate our findings for other microgel systems. As already mentioned above in the key changes, we now extend our study with two additional sets of microgels, which allows us to obtain an in-depth resolution to the architecture and morphology of the stabilizing microgels to their corresponding emulsion behaviour.

In addition, we now provide a detailed characterisation of all used microgels including hydrodynamic diameter vs temperature, swelling ratio and compressibility at the liquid interface in Supplementary Figure S1 (added below for clarity). This demonstrates the gradual change in bulk and interfacial properties of the microgels used in this study. The elastic modulus of the microgels used in this study have already been determined in previous work using liquid phase atomic force microscopy. Kühnhammer et al. (Ref. 54 in our manuscript) determined the Young's Modulus of regular microgels to 100 kPa for microgels with 2 mol% crosslinker, gradually increasing up to 400 kPa for microgels with 10 mol% crosslinker. The Young's Modulus of core-shell to hollow microgel system have been measured by Vialetto et al. (Ref. 68 in our manuscript) to 100 kPa for core-shell microgels, 35 kPa for partially degraded core-shell microgels and 10 kPa for hollow microgels. Since we used the same synthesis protocol in our study, we expect elastic modulus to be similar.

We added the following paragraph to the main text:

We tune the softness of the stabilizing microgels by varying their crosslinking densities, which is known to systematically tune their swelling (Fig. S1a,b),⁵³ elasticity⁵³⁻⁵⁵ and compressibility at liquid interfaces (Fig. S1c).^{53,56} We compare the full polymer-to-colloid range (Fig. 2a), starting from linear polymer via ULC microgels towards regular microgels with increasing crosslinking densities, which approach a colloid-like behaviour.

It had been shown previously that G' varies significantly with temperature for the thermo-responsive hydrogels.

Indeed, the mechanical properties of bulk thermo-responsive microgel suspensions change with increasing temperature, and it has been widely reported that the collapse of the microgels induces a

change in packing fraction, which can lead to fluidisation (Senff, H., and W. Richtering. *The Journal of chemical physics* (1999) Ref 47, in our manuscript). Interestingly, and maybe at first glance a bit counterintuitively, no fluidisation occurs in a microgel monolayer confined at a liquid interface (Figure 1). We infer that the interfacial energy gained due to the microgel adsorption at the liquid interface counteracts the lateral shrinkage of the microgels and only vertical shrinkage is possible. We now note that such behaviour is initially counterintuitive, as fluidisation of the interface due to reduced area would be “comparable to bulk fluidisation with reduced volume fraction.

We added the following sentence to the main text:

Previously proposed mechanisms for stimuli-responsive destabilisation would lead to either a fluidised interface ($G^{s'} \ll G^{s''}$) due to lower surface coverage (Fig. 1a (i)-(ii)), comparable to bulk fluidisation with reduced volume fraction,⁴⁷ or a weaker interface due to aggregation (Fig. 1a (iii)).

Further, to validate parameters choice in the computational modeling of microgels, I would expect reproducing volume phase transition in microgels with temperature change. In other words, reproducing experimental plot provided in Figure S1, at least with a few simulation points.

We have updated Supplementary Figure 1, which now contains the characterization of all synthesized and in-silico microgels used in this study (added below for clarity). In particular, the swelling of in-silico microgels is tuned via the α parameter, that governs the attraction between the monomer units of the microgels. $\alpha = 0$ corresponds to the swollen state, while $\alpha = 1$ corresponds to the collapsed state. In Supplementary Figure S1, we provide plots of the radius of gyration vs α , swelling ratio β vs α and snapshots of the bulk microgel morphologies for $\alpha=0$ and $\alpha=1$.

Figure S1: Bulk and interfacial characterisation of the microgels used in this study. a-c) Series of microgels with increasing crosslinker content from ultra-low crosslinked (ULC) microgels to regular microgels with 1 mol% up to 10 mol% crosslinker. a,b) Hydrodynamic diameter D_H (a) and swelling ratio β (b) as a function of temperature. c) Relative area vs surface pressure measured on a Langmuir trough. The area at a surface pressure of 30 mN/m has been defined as 100 %. d-e) Hydrodynamic diameter D_H (d) and swelling ratio β (e) as a function of temperature for core-shell microgels with systematic core degradation towards hollow microgels. f) Relative area vs surface pressure. g-i) Characterisation of regular and ULC in-silico microgels. g) Radius of gyration (R_G) vs α . h). Swelling ratio β vs α . i) Corresponding snapshots in bulk water at $\alpha=0$ and $\alpha=1$.

Minor comment: double-corona could be more clearly defined somewhat earlier in the text.

As mentioned in the key changes above, we now refer to double-corona microgels as bridging microgels to avoid confusion.

Reviewer #2 (Remarks to the Author):

The manuscript by Rey et al reports a very interesting study on the stability of emulsions stabilized by temperature-sensitive PNIPAM microgels. They investigate properties of interfaces and emulsions using different experimental and computational techniques. They argue that the presence of microgels that are attached to the water-oil interface of two different droplets is required to have a temperature dependent (in)-stability of emulsions. I fully agree with the authors that the stimulus-dependent stability of emulsions and thus the coalescence of microgel-covered emulsion droplets is still an open question. I also agree that understanding the (in)stability of such emulsions is important for exploiting the stimulus-sensitivity of such emulsions. Thus, I think the ms will be suitable for publication in this journal.

We thank the Reviewer for their positive assessment of our manuscript and for the helpful comments, which we address below. To more clearly structure the description of our changes in the answer letter, we answer some of Referee 2's more substantial comments first.

However, I have several comments which - in my view - suggest / require major revisions.

The authors use the term "double-corona particle morphology". I find this term very misleading: at least to me, when reading the title and the text, I interpreted this term such that it indicated that the microgels have intrinsically a special "double-corona particle morphology" (i.e. even in bulk). This is, however, not the case. The authors use normal PNIPAM microgels, which have in solution a core-corona type morphology and which also leads to a core-corona (or fried-egg) like structure when adsorbed at an oil-water interface. What - I think - the authors want to express is that it is important that a microgel is adsorbed to the surface of two different droplets. It is this bridging, that is relevant. Of course, soft microgels will then have the core-corona (or fried-egg) like structure at both interfaces, but I would not call this "double-corona". I suggest the authors skip the term "double-corona particle morphology" and rather speak of bridging microgels.

We thank referee 2 for pointing out that the term "double-corona particle morphology" is misleading and not intuitive. As mentioned in the key changes above, we now refer to double-corona microgels as bridging microgels to avoid confusion.

The authors claim, that emulsions where the droplets are stabilized by a (mono)layer of microgels and the where the droplets are separated, do not break at high temperature. Well I accept that the authors made this observation, but I have two concerns:

(i) I feel that this is a rather artificial state. Real emulsions will be subject to "mechanical" disturbance, be it shaking, flow etc. I would be very surprised if these emulsions will be stable under such conditions.

Referee 2 correctly points out that mechanical disturbances, e.g., via shaking, can compromise emulsion stability. We added a detailed discussion to the Supplementary Information, which reads as:

Emulsion stability under mechanical disturbances:

To estimate the influence of mechanical disturbances, we compared the behaviour of flocculated and dispersed emulsions under shaking, both at room temperature and at 55 °C. We placed the emulsion vials flat on an incubator shaking plate, which allows us to simultaneously heat and shake. The incubator shaking plate moved in a circle with a diameter of 2 cm and we tested two shaking parameters: 60 rpm and 150 rpm. At 60 rpm, the emulsion was gently shaken back and forth within

the vial, while at 150 rpm the flow within the vial was disruptive and chaotic. We compare the emulsions before and after 4 hours of treatment in Fig. S7.

We observe the following behaviour for dispersed emulsions. Under gentle shaking at 60 rpm the dispersed emulsions remain stable at both room temperature and when heated. The stability of the dispersed emulsion under mechanical disturbance, which brings the droplets into contact, above the T_{VPT} supports our conclusions in the main manuscript, *i.e.* the stability is not an artefact of the quiescent conditions keeping the droplets separated. At higher shaking speed (150 rpm), the dispersed emulsions undergo shear-induced coalescence at both temperatures (clear oil layer on top). Such processes are widely reported, with the stabilisation mechanism being disrupted as the droplets are sheared past one another.⁵ Previously, stronger interfaces have been found to resist such shear-induced coalescence, *e.g.*, flocculated *vs* dispersed fumed silica.⁶ This suggests that the shear forces at high shaking speeds are sufficient to yield the interfacial monolayer, as in Fig. 1. An order of magnitude dimensional analysis estimate from the interfacial yield strength ($\sigma_y \sim 10^{-4}$ Pa.m) and droplet size (10 μm) suggests stresses of ~ 100 Pa to be sufficient to yield all interfaces consistent with the forces from shaking.

In contrast, for flocculated emulsions shear-induced coalescence is found at both shaking speeds above and below the T_{VPT} . We infer that the shear forces acting on the flocculated emulsions are sufficient to break the bridged region of two flocculated droplets in a “zipper-like” manner.⁷⁻⁹ To summarize, these experiments demonstrated that dispersed emulsions remain stable under mild shaking at 55 °C, but can be broken even at room temperature under high shear.

In addition, we added Figure S7

Figure S7: Simultaneous shaking and heating of dispersed and flocculated emulsions in an incubator. The incubator shaking plate moved in a circle with a diameter of 2 cm at 60 rpm or 150 rpm and the temperature is set to either 22 °C or 55 °C. At 60 rpm, the emulsion was gently shaken back and forth within the vial, while at 150 rpm the flow within the vial was disruptive and chaotic. The vials before (left) and after 4 hours of shaking/heating (right) are shown next to each other for comparison.

(ii) *This observation is different from what has been reported in the literature and I do not think that the authors' observation can be generalized.*

In our recent study (Petrunin, A. V.; Bochenek, S.; Richtering, W.; Scotti, A. Harnessing the Polymer-Particle Duality of Ultra-Soft Nanogels to Stabilise Smart Emulsions. Phys Chem Chem Phys 2022. <https://doi.org/10.1039/d2cp02700c>.) we found a behavior that is different from the observations by Rey et al. With ultra-low crosslinked (ULC) microgels one can prepare well dispersed emulsions (no flocculated droplets) but the emulsions break at high temperature. Also, emulsions prepared with microgels containing 1% BIS, non-flocculated droplets can coalesce.

In one of our earlier studies, we found that emulsions can brake under conditions where the microgels are still colloidally stable in the aqueous phase or when the microgels flocculate in the aqueous phase (Wiese, S.; Tsvetkova, Y.; Daleiden, N. J. E.; Spiess, A. C.; Richtering, W. Microgel Stabilized Emulsions: Breaking on Demand. Colloids and Surfaces A 2016, 495, 193–199.)

We thank Referee 2 for this very helpful comment. Indeed, the recent work by Petunin et al. clearly demonstrates that dispersed emulsions stabilized by ULC microgels or regular microgels with low crosslinking densities destabilize upon temperature increase, a behaviour we had overlooked during initial screening prior to this study. In the revised manuscript, we address the responsive emulsion behaviour reported by Petunin et al. and discuss how the responsiveness of dispersed emulsions is linked to the morphology of their stabilizing microgels. We extended our study with two additional sets of microgels (variation in crosslinking density, core-shell towards hollow microgels) to cover the wide range tuneable physical-chemical parameters in soft microgel systems, which we describe in the key changes above. Our study now provides a complete picture on how the emulsion stability and stimuli-responsiveness is coupled to the stabilizing microgel morphologies.

I have the impression that the size of the bridging microgel orthogonal to the interface, which changes with temperature is relevant for the onset of coalescence.

Referee 2 correctly predicted that the onset and the duration of the coalescence is affected by the size and polymer density of the bridging microgels. In an additional experiment, we compare the onset of coalescence for flocculated emulsions stabilized by microgels with different crosslinking densities. Since softer microgels have a lower polymer content (Rey et al., Ref. 78 in our manuscript) and are able to deform more at liquid interfaces (Fig. S1c), the size and polymer density of the microgels orthogonal to the interface increases with increasing crosslinking density. Here, we place the flocculated emulsion vials in an incubator preheated to 55 °C and qualitatively compare the state of the emulsions at different time steps while the emulsions heat up (Fig. S8). We qualitatively observe a delay in the onset of coalescence from emulsions stabilized by low crosslinked microgels towards emulsions stabilized by microgels with higher crosslinking density.

And we added Figure S8 to the Supplementary Information:

Figure S8: Time series of dodecane in water emulsions in an incubator at 55 °C. Flocculated emulsions stabilized by lower crosslinked microgels break more rapidly compared to the ones stabilized with higher crosslinked microgels, whereas the dispersed emulsion remains stable.

Can the authors evaluate the cryo-SEM images in more detail and determine e.g. the surface coverage for different cases?

Furthermore, I suggest that the authors also include a discussion of their results in comparison to: Destribats, M.; Eyharts, M.; Lapeyre, V.; Sellier, E.; Varga, I.; Ravaine, V.; Schmitt, V. *Impact of pNIPAM Microgel Size on Its Ability To Stabilize Pickering Emulsions*. *Langmuir* 2014, 30 (7), 1768–1777. I think their observation of size effects fits nicely to the results obtained by Rey et al. I also suggest discussing the images in Figure 2 to the images reported by Destribats et al. *Langmuir* 28, 37744 (2012).

We added a short discussion in the main text combined with an in-depth discussion on the microgel assembly at emulsion interfaces in the Supporting Information. We first compare the surface coverage at emulsion interfaces to our previous studies on Langmuir interfaces using the same microgel particle system. Then, we qualitatively compare the assembly to previous work by Destribats et al. and discuss how emulsification parameters can influence the interfacial assembly. We added the following Chapter to the Supporting Information, which reads as:

Microgel assembly at emulsion interfaces:

In the following, we discuss the assembly and packing density of microgels (5 mol% crosslinker) stabilizing oil in water emulsions. The interfacial assembly of the microgels in this study have been the subject of prior self-assembly investigations using a Langmuir-Blodgett trough,¹⁰ where the interfacial layer was deposited onto a substrate with the assembly analysed ex situ. At low compression or high area per particle, they assemble into a hexagonal non-close packed arrangement where the microgels are in corona-corona contact.^{10–12} Upon further compression, the microgels undergo an isostructural solid-solid phase transition to a hexagonal close packed phase.^{10–12} Interestingly, a recent study using small-angle light scattering could not find any evidence of an isostructural phase transition at the liquid interface.¹³ Instead, the isostructural phase transition

seems to be a result of the microgel's adhesion to the solid substrate and immersion capillary forces that occur upon drying.¹³

Here, we prepare emulsions with different microgel concentrations and investigate the packing density and assembly of the stabilizing microgels at the droplet interface using cryo-SEM (Fig. S9). We find a decrease in area per particle (A_p) with increasing microgel concentrations. Further, the microgel assembly at the droplet interface is qualitatively similar compared to our previous study of flat Langmuir monolayers.¹⁰ For low microgel concentrations (0.15 wt%), the microgels are in a hexagonal non-close packed arrangement where they are in corona-corona contact (Fig. S9a). For intermediate microgel concentrations (0.3 wt%, 0.6 wt%), we find a coexistence of a hexagonal non-close packed and a hexagonal close packed phase (Fig. S9b,c), which would correspond to the isostructural phase transition. At higher concentrations (1.2 wt%, 1.5 wt%), the assembly is hexagonal close packed (Fig. S9d,e). Our data suggests that the isostructural phase transition occurs at the liquid droplet interfaces even in the absence of capillary forces upon drying.

With an increase in microgel concentration, the degree of flocculation decreased. In flocculated emulsions (Fig. S9a-d), we found non-close packed and close packed microgel assemblies. Noteworthy, for 1.2 wt% microgels, most of the emulsion is dispersed but some bridging points in flocculated emulsions can still be observed despite the close packed arrangement. In addition, we find that the microgel assembly in the proximity of bridging points is distorted as the monolayer of the neighbouring droplets interfere with the assembly due to their proximity. The apparent area per particle thus appears higher. Importantly, all investigated dispersed emulsions are characterized by a close packed microgel monolayer (Fig. S9e), while non-close packed assemblies were only observed in flocculated emulsions (Fig. S9a-c). These findings qualitatively agree with a study by Destribats et al., who tuned the packing density via the emulsification temperature instead of microgel concentration.⁸ Similarly, they report dispersed emulsions for close packed microgel monolayers and flocculated emulsions for non-close packed monolayers.⁸

We should note that controlling the microgel assembly at emulsion interfaces is much more challenging compared to frequently used assemblies in a Langmuir trough. In emulsions, the microgel assembly depends on many different experimental parameters, such as the concentration of microgels in the aqueous phase (as shown above), their surface to volume ratio, volume concentration adsorption rate, ability to spread at liquid interfaces, size and mobility, swelling ratio and also the emulsification method (*e.g.* shaking, vortex mixing, rotor stator), energy input and oil/water ratio. The detailed influence of some of the mentioned parameters have been elaborated in previous work.^{8,9,14,15} The parameters mentioned above are coupled. For example, previous work showed that increasing either microgels crosslinking density or size while keeping a fixed concentration leads to a decrease in packing density at the emulsion interface and an increase in the degree of flocculation.^{8,14} In order to qualitatively compare either dispersed or flocculated emulsions (Fig. 2), we increase the microgel concentration in the aqueous phase with increasing crosslinking density while keeping the emulsification parameters constant.

And we added Figure S9:

Figure S9: Cryo-SEM images of dodecane in water emulsions stabilized by 5 mol% PNIPAM microgels. The emulsions are prepared using different concentrations of microgels in the aqueous phase: a) 0.15 wt%, b) 0.3 wt%, c) 0.6 wt%, d) 1.2 wt%, e) 1.5 wt%. With increasing microgel concentrations, we find a decrease in area per particle (A_p). In addition, we observe an iso-structural solid-solid phase transition from a hexagonal non-close packed phase (a) via a phase transition region (b,c) to a close packed phase (d,e), which qualitatively agrees with previous studies on Langmuir monolayers.¹⁰ With increase in microgel concentration, the degree of flocculation decreased. d) For 1.2 wt% microgels, most of the emulsion is dispersed but some bridging points in flocculated emulsions can still be observed despite the close packed arrangement. In addition, we find that the microgel assembly in the proximity of bridging points is distorted as the monolayer of the neighbouring droplet interfere with the assembly due to their proximity. The apparent area per particle thus appears higher. Importantly, all investigated dispersed emulsions are characterized by a close packed microgel monolayer (e), while non-close packed assemblies are only present in flocculated emulsions (a-c). Scale bars: 2 μm .

In the main text, we added the following sentence with regards to bridging microgels in flocculated emulsions:

Previous work also reported a combination of mono- and bilayers within bridging points for softer microgels.⁵⁸ We infer that the bridging microgels still act as weak links, inducing coalescence.

Concerning the simulations, am I right that the simulations did not include dispersion forces between the oil phases? I do not expect that this changes the conclusions, but I wonder why this has not been taken into account?

Yes, the referee is right that we have neglected the dispersion forces between the oil phases. These forces contribute to the osmotic pressure as $\Pi = -A / (6 \pi d^3)$ where A is the Hamaker constant and d the distance between two flat oil interfaces. In the units chosen in Figure 4, the contribution is less than 1 percent even at the smallest distance d if a typical value of $A = 10^{-20}$ J is taken for the Hamaker constant. This means that dispersion forces are indeed negligible as conjectured by the referee.

We added the following sentences in the method section:

Dispersion forces between the two oil phases are negligible. They contribute to the osmotic pressure as $\Pi = -A / (6 \pi d^3)$ where A is the Hamaker constant and d the distance between two flat oil interfaces. In the units chosen in Figure 4, the contribution is less than 1 percent even at the smallest distance d if a typical value of $A = 10^{-20}$ J is taken for the Hamaker constant.

Concerning Figure 3 c/e: I guess, I understand how the authors calculate the osmotic pressure, but I have a (may be naive) question. Once the microgel is completely collapsed, I expect that the excluded volume of the polymer segments leads to a repulsion between the interfaces, and coalescence is not possible as the microgel cannot be compressed further. In order to have coalescence, the microgel must be pushed out. Am I right?

The referee is right. In our simulations, we confine the microgels in cylindrical box (Wigner-Seitz cell) to mimic the hexagonal arrangement within microgel monolayers. Therefore, they cannot be pushed out of the interface. We apologize for the previously unfavourable representation in Fig. 4a, which we now updated to illustrate the lateral confinement by illustrating the Wigner-Seitz cell in Fig. 4a,d,e, which now reads as:

Figure 4: Monomer-resolved Brownian dynamics simulations of regular thermo-responsive microgels confined between two planar liquid interfaces confined in a Wigner-Seitz cell. a) Schematic illustration of the simulation set-up (top) and the corresponding interfacial monomer potential (bottom). The monomers are shown as green/purple spheres and the crosslinker as grey spheres. The potential of the oil phase was varied (black, dark grey and grey curves) to allow more beads (size σ) in the oil phase, mimicking changes in microgel wettability. b,c) Osmotic pressure Π exerted by the microgel onto the liquid interface as a function of distance d between the two liquid interfaces of one microgel adsorbed to both interfaces (filled, representing flocculated emulsions) and two confined microgels each adsorbed to one interface (hollow, representing dispersed emulsions) in the swollen (b) and in the collapsed (c) state. A positive Π corresponds to a repulsive force of the microgels onto the two liquid interfaces. d,e) Representative snapshots of the microgels at different compression states in the swollen (d) and collapsed (e) state. For one microgel, we notice the formation of a catenoid structure upon collapsing.

The referee further correctly predicted that, since the microgels in our simulations are not able to be pushed out of the interface, the simulations no longer reflect the experimental system and the excluded volume of the polymer segments eventually leads to a repulsion between the interfaces when the distance between them approaches zero. At such close distances, however, the two interfacial potentials are overlapping and an exchange between the two adjacent oil droplets is to be expected. We agree with the referee that the simulations are suitable to measure the osmotic pressure acting on the liquid interfaces, which helps us understand whether coalescence would take place or whether the microgels provide sufficient stabilisation to prevent coalescence. However, since the simulation does not allow the microgels to escape laterally, the mechanism on how the microgel monolayer fails during coalescence is not revealed.

We added the following paragraph in the discussion, how coalescence events are linked to potential microgel desorption and interfacial wrinkling.

Upon coalescence, the interfacial area of the droplets is reduced leading to a lateral compression of the interface and additional effects may arise controlling the rate of coalescence. The shrinking area will eventually lead to failure of the microgel monolayer via either desorption, interfacial wrinkling, or the formation of multilayers. While these larger scale structural effects are not probed by our Brownian dynamics simulations, post-coalescence imaging can indicate the possible mechanism in action. Soft microgels are known to be able to desorb from the liquid interface to the aqueous phase upon lateral compression.⁷² This may cause the clear separation into an aqueous and oil phase after de-emulsification (Fig. 2a). On the other hand, interfacial wrinkling has been observed in Langmuir monolayers of regular microgels with higher crosslinking densities.⁷³ This may explain the formation of a microgel cluster, which after de-emulsification sits between the aqueous and oil phases (Fig. 2a, Fig. S13). Confocal microscopy of said cluster reveals the microgel shells of the former emulsion droplets remain relatively intact even after the encapsulated liquid is drained (Fig. S13). Seemingly, redispersion of the microgels into the aqueous phase is hindered. In addition, some oil droplets are found in the cluster, which likely became trapped during the de-emulsification process, leading to it remaining buoyant between the bulk oil and aqueous phases. To summarize, the tendency of more crosslinked microgels towards interfacial wrinkling instead of desorption may slow de-emulsification of flocculated emulsions compared to softer microgels (Fig. S8). In addition, monolayer failure through wrinkling may likely be an additional component that hinders coalescence, promoting the observed stable dispersed emulsions for more crosslinked microgels. This suggests that the microgel architecture and morphology affect coalescence dynamics in parallel to the effects initiating or preventing coalescence.

We added the following Figure to the Supporting Information:

Figure S13: Confocal microscopy reveals the microstructure of a microgel cluster resulting after de-emulsification of flocculated emulsions. a,b) After de-emulsification, microgels stabilizing flocculated emulsions in part aggregate into a cluster, which sits between the oil and water phase. c,d) Confocal microscopy after transferring parts of the cluster to a cover slip. Note: oil may be drained from the cluster after transfer. Confocal microscopy reveals that the cluster consists of a joint 3D microgel network which persists after droplet coalescence and drainage of the encapsulated oil. We infer that the microgels turn fluorescent due to non-covalent binding of Nile red dye,¹⁷ which we use to visualize the oil phase. e) In addition, some oil droplets are found in the cluster, which likely became trapped during the de-emulsification process, leading to it remaining buoyant between the bulk oil and aqueous phases. The 3D microgel network within the cluster after coalescence indicates that a significant amount of the stabilizing microgels is not able to desorb from the liquid interface and coalescence may instead lead to buckling of the microgel monolayer. Scale bars: 50 μm .

I agree with the authors' statement "that the vertical structure of microgel stabilizers, ..., is of vital importance". This agrees with our recent report by Petrunin et al in PCCP 2022.) where we compare normal PNIPAM microgels, ultra-low crosslinked (ULC) microgels and linear polymer. However, while I see the relevance of vertical structure and the presence of bridging, I have the impression that the situation

is still more complex. I wonder if the authors would agree to this view on stimuli-sensitive microgel stabilized emulsions:

In order to have a stable emulsion one needs: separated droplets, covered by a dense monolayer of microgels and a vertical (orthogonal) size of the microgel that provides sufficient steric stabilization. That suggests a minimum size of the microgels. Keeping in mind that big microgels are slow, it will be difficult to avoid having some microgels adsorbed to two droplets (bridging) during formation of the emulsion. The coalescence of droplets requires close contact of the oil-water interfaces. Here bridging microgels will play an important role as their vertical (orthogonal) size (which depends on temperature as well as the density of polymer segments) will set the relevant length scale for separating the oil-water interfaces. Finally, mobility of the microgels at the interface, the viscoelasticity of the microgel covered interface and the rate of microgel desorption upon lateral compression of the interface are important for the transition from dispersed to aggregated (bridged) droplets and the rate of coalescence of bridged droplets.

I do agree with the authors' statement that experiments at flat monolayers showed that microgels mainly shrink orthogonally and do not leave the interface. However, when emulsion droplets coalesce and finally macroscopic phase separation occurs, microgels have to leave the interface. It has also been observed that microgels leave the interface when the monolayer is compressed in a Langmuir trough.

We agree with Referee 2 on his summary on stimuli-sensitive microgel stabilized emulsions. Stable dispersed emulsions are obtained when emulsified with sufficient microgels to form a dense monolayer. Low shear emulsification with an excess of microgels favours the formation of dispersed emulsions and minimizes the possibility of bridging emulsion droplets. Bigger microgels or highly crosslinked microgels require a higher amount of microgels in the aqueous phase to counteract the slower diffusion. In addition, the revised manuscript we discuss in-depth with the new sets of microgels that if the stabilizing microgels have characteristic core-corona morphology with a core extending into the aqueous phase, they can provide sufficient steric stabilisation even when heated above T_{VPT} . If the microgels contain little crosslinker, they deform into a flattened morphology at the liquid interface similar to a polymer film, which does not provide sufficient stabilisation above T_{VPT} and macroscopic coalescence is observed. In flocculated emulsions, the microgels adsorbed to two liquid interfaces are identified as weak links, inducing an attractive osmotic pressure onto neighbouring interfaces above T_{VPT} , inducing coalescence.

We added the following paragraph summarizing the behaviour of dispersed emulsions in the discussion:

In dispersed emulsions, the response depends on the architecture and interfacial morphology of the stabilizing microgels. Stimuli-responsive emulsions are obtained for low crosslinked or hollow microgels. Like linear polymers, they extend at the liquid interface and assume a flattened pancake morphology. While these emulsions are stable at room temperature, the flattened microgels collapse into a thin film above T_{VPT} , which is not sufficient to prevent coalescence. Brownian dynamics simulations show a negative osmotic pressure, i.e. attractive force, even for two microgels. By contrast, dispersed emulsions stabilized by more highly crosslinked colloid-like microgels do not coalesce upon temperature increase. We attribute the absence of stimuli-responsiveness to their characteristic core-corona morphology with a core extending into the water phase, creating a repulsive force between interfaces, even above T_{VPT} . To summarize, we find that the occurrence and absence of a thermo-responsive emulsion breaking mechanism is, therefore, linked to the morphology of the stabilizing microgels and not the interfacial properties of the assembly.

In addition, we also discuss how microgels need to be designed in order to obtain long-term stable, yet stimuli-responsive emulsions. We added the following paragraph in the discussion section:

Our investigations highlight the importance of the nature of the stabilizing microgel particles, i.e. more polymeric vs more colloidal. The microgels' colloidal properties are fundamental to the long-term stability of Pickering emulsions. Emulsions stabilized by linear polymers are not long-term stable (Fig. S5), which we attribute to the weak interface (Fig. S4) as shown in recent work on dendronized polymers.⁷⁴ On the other hand, non-responsive emulsions are obtained if the stabilizing microgels are too hard with a limited deformability. At the liquid interface, such microgels assume a core-corona morphology with a core extending into the aqueous phase that provides sufficient steric stability even above T_{VPT} . In our study, we find that ULC microgels, regular microgels with 1 mol% crosslinker and hollow microgels fulfil the balance between polymeric and colloidal properties required to enable stimuli-responsive emulsions. Their colloidal properties lead to an elastic interface providing long-term stability while their polymeric properties allow for them to spread and flatten at the liquid interface, enabling stimuli-responsiveness. On the other hand, microgels adsorbed to both droplets in flocculated emulsions enable stimuli-responsiveness regardless of their internal architecture. We believe that the importance of the microgel interfacial morphology and their polymer-colloid duality on the stimuli-responsive emulsion behaviour will also be of relevance to other stimuli-responsive systems. The presence of bridging microgels has been reported within thermo-responsive foams,^{54,75} which may similarly serve as weak links prompting the macroscopic foam destabilisation. Alternatively, emulsions¹⁴ and foams⁷⁶ stabilized by pH-responsive microgels can be broken on demand upon change in pH. Similarly, we expect that in those systems the colloidal properties will ensure the stability while the polymeric properties and softness of the microgels enable the responsive behaviour.

With regards to microgels leaving the interface upon coalescence, we believe that this only occurs for softer microgels, whereas harder microgel interfaces likely fail via interfacial wrinkling. We addressed this aspect in the answer to the previous reviewer comment above.

In this context, I also find the term "quasi-neutral PNIPAM microgel" misleading. I fully agree, that the microgels do not carry many charges, but the small number of charges make them colloidally stable at $T > VPTT$ (as they are prepared in surfactant-free precipitation polymerisation). I suggest that the authors do some further control experiments in the presence of salt at a salt concentration where the microgels flocculate when heated above the $VPTT$.

We agree with the referee and apologize for using the term "quasi-neutral". The few charges on the microgels make them colloidally stable above T_{VPT} and prevent the stabilized emulsion droplets from flocculating. We have now removed the term "quasi-neutral" from the manuscript. In addition, we performed a control experiment like the one in Fig. 2c but at 25 mM NaCl in the aqueous phase. At 55 °C, the microgels in the aqueous phase aggregate into a gel and the dispersed emulsion droplets flocculated and became arrested. Partial de-emulsification occurred over 24 hours at 55 °C, highlighting the importance of the small number of charges on microgels in preventing coalescence. A detailed insight in how salt affects the destabilisation mechanism though is challenging as the formation of microgel multilayers due to attractive forces between them will have to be considered, which are difficult to replicate in model interfaces we used for interfacial rheology. We thus believe that the aspect of salt on the destabilisation mechanism is beyond the scope of this study and should be the subject of follow-up research.

As further comments:

*I think it makes sense to compare the bridging behavior of microgels attached to two oil surfaces with the properties of foams. There are reports e.g. by the Regine von Klitzing group on foams stabilized by PNIPAM microgels (e.g. *Soft Matter* 2022 <https://pubs.rsc.org/en/content/articlehtml/2022/sm/d2sm01021f>.) (I think there are more papers on that topic in the literature). In thin films, the microgel can also be attached to 2 interfaces and bridges these interfaces.*

We added a comparison to foams stabilized by PNIPAM microgels in the discussion, which now reads as:

... We believe that the importance of the microgel interfacial morphology and their polymer-colloid duality on the stimuli-responsive emulsion behaviour will also be of relevance to other stimuli-responsive systems. The presence of bridging microgels has been reported within thermo-responsive foams,^{54,75} which may similarly serve as weak links prompting the macroscopic foam destabilisation. Alternatively, emulsions¹⁴ and foams⁷⁶ stabilized by pH-responsive microgels can be broken on demand upon change in pH. Similarly, we expect that in those systems the colloidal properties will ensure the stability while the polymeric properties and softness of the microgels enable the responsive behaviour.

*As a final (very minor) note concerning liquid phase colloidal probe AFM experiments suggested by the authors in the last paragraph. We (or rather the Butt group) tried that (but with a hard colloidal probe, not covered by microgels) (Bochenek, S.; McNamee, C. E.; Kappl, M.; Butt, H.-J.; Richtering, W. *Interactions between a Responsive Microgel Monolayer and a Rigid Colloid: From Soft to Hard Interfaces. Phys Chem Chem Phys* 2021, 23, 16754–16766.) These are tricky experiments and challenging to be interpreted...*

We thank the referee for the suggested reference, which we now included in the manuscript.

Reviewer #3 (Remarks to the Author):

The article of Rey and coworkers addresses the stability of emulsions stabilized by microgel particles. They claim to have unraveled the mechanism that determines the stability of these exciting formulations. Namely, they associate the increased stability with a double Corona particle morphology that enables stimuli-responsive emulsions.

This article addresses an interesting and timely question that appears to be unanswered today. Unfortunately, the paper also has several shortcomings that make it unsuitable for publication in Nature Communications at the current stage.

We thank referee 3 for pointing out that our manuscript addresses an interesting and timely question and encouraging us to make our figures more intuitive. We have addressed all raised concerns (details below) and hope that we can convince referee 3 to consider a positive evaluation of our manuscript.

First and most importantly, the authors claim that the double-corona particle morphology (of microgels) is a crucial aspect leading to stable emulsions. The authors, however, need to explain what they mean by a double Corona particle morphology (?), ideally adding a sketch. The paper is vague about these questions, and the figures reveal little. For example, in Figure 2, many panels from cryo-SEM are added, but only in two of those some color patches are added. Again, I need clarification on what they show or proof. Also, here, a sketch of the drawing of microgel-morphology would be helpful.

We apologize for the non-intuitive term “double-corona morphology”. As mentioned in the key changes above, we now refer to double-corona microgels as bridging microgels to avoid confusion.

In addition, for each type of microgel investigated with cryo-SEM we added sketches, Fig. 3 (former Fig. 2) a,f,k ,Fig. 5f and Fig. 6a,f,k, of the respective microgel morphologies, which will help the readers spot the difference in microgel morphologies in the original cryo-SEM images. A detailed discussion on how the microgel morphology is linked to the forces they exert onto the liquid interfaces is provided in the key changes above.

As a reviewer, I would appreciate a better-quality copy of the manuscript. For a review, the font is small and inconvenient to read. One should certainly improve the figures. The fonts e.g., in Figure 1 and plots, are tiny. The images of the vials in Fig 1 c were taken at an oblique angle, and the phase separation is hard to see.

We have now changed the font size of the manuscript to 12 pt, which should be more convenient to read. We also adjusted the document borders so that the figures are now displayed in the size according to the *Nature Communications* template.

We agree with Referee 3 that Fig. 1 was too crowded, and the font size was too small. In the revised version, we split former Fig. 1 into a rheology figure and an emulsion stability figure to provide a more intuitive structure to each figure. Further, we made sure that the fonts in the figures are of sufficient size.

We thank Referee 3 for pointing out that the phase separation within the vials (former Fig. 1c) is hard to see due to the oblique angle. In the revised version, we retook all images of vials with a photo studio. Hereby, all vials are photographed straight on in front of a dark background. This helps visualizing the magnitude of the phase separation.

The changes made to former Fig. 1, now Fig. 1 and Fig. 2 are shown below.

Figure 1: Interfacial response of thermo-responsive PNIPAM microgels. a) Previously proposed destabilisation mechanisms: covered interface to fluidised interface with lower coverage due to (i) shrinkage, (ii) desorption, or (iii) weakening of the interface due to aggregation. b) Interfacial shear rheology with double wall ring geometry. c-j) Interfacial rheological response to changing temperature: c-f) Oscillatory strain amplitude sweep for (c) ultra-low crosslinked (ULC) microgels at a frequency $f = 0.1$ Hz, microgels with (d) 1 mol% crosslinker, (e) 5 mol% crosslinker and (f) 10 mol% crosslinker at $f = 0.2$ Hz. Storage (G^s , filled) and loss (G^{s*} , open) moduli with strain amplitude, γ_0 , at low temperature, $T < T_{VPT}$ (blue), and high temperature, $T > T_{VPT}$ (red). Shading, resolution limit.⁵² g-j) Linear viscoelastic response with increasing T : (g) at $\gamma_0 = 0.05$, (h-j) at $\gamma_0 = 0.01$.

Figure 2: Stimuli-responsive behaviour of dodecane in water emulsions stabilized by PNIPAM microgels. a) Vials of dispersed (top) and flocculated (bottom) emulsions stabilized by linear PNIPAM and PNIPAM microgels with increasing crosslinking densities prepared at 22 °C (left) and after storage at 55 °C for 4 hours (right). The emulsions show creaming due to the density mismatch between dodecane and water. No flocculated emulsions were obtainable for linear PNIPAM. b-d) Optical microscopy images as a function of temperature of dispersed emulsions, stabilized by (b) ultra-low crosslinked (ULC) microgels and (c) 5 mol% microgels, and (d) flocculated emulsions, stabilized by 5 mol% microgels. We classify the emulsion behaviour into three regimes. Dispersed emulsions stabilized by low crosslinked microgels, or linear polymers are responsive and break above T_{VPT} (purple frame). Dispersed emulsions stabilized by microgels with higher crosslinking densities remain stable and most of the droplets do not coalesce even up to 80 °C (red frame). All flocculated emulsions, on the other hand, destabilize above T_{VPT} (orange frame). Scale bars: 50 μm .

Figure 2 looks like a random collection of SEM images where, as mentioned before, only three small parts are highlighted with red color. The connection between the simulations shown in Figure 3 and the (yet-to-be-defined) double-corona morphology is difficult to grasp.

As previously mentioned above, we made several adjustments to Fig. 3 (former Fig. 2) to make it more intuitive to read. We added additional sketches for all microgel morphologies depicted in the cryo-SEM images (Fig. 3a,f,k), which will guide the readers. Further, we want to emphasize the motivation behind Fig. 3 and explain why it is not just a “random collection of SEM images”. In Fig. 3, we compare three types of emulsions stabilized by the same microgels. Fig. 3a-e shows cryo-SEM images of flocculated emulsions. Fig. 3f-j are dispersed emulsions frozen with liquid nitrogen

from room temperature and Fig. 3k-o the same dispersion frozen with liquid nitrogen from 55 °C. The cryo-SEM investigation will now highlight that even though all emulsions are stabilized by the same microgels, the corresponding microgel morphology differs. To emphasize the difference between the investigated emulsions we added sketches of the microgel morphologies and an illustration of a thermometer.

Next, the reason why we show 4 cryo-SEM images each is due to the different magnifications. In the second column of Fig. 3 (b,g,l), we show the cryo-SEM images on the emulsion level. This reveals the existence of bridging points for flocculated emulsions, while dispersed emulsions are characterized by a spherical shape and a complete microgel monolayer. In the third column (Fig. 3 c,h,m) we zoom onto the microgel monolayer, which reveals the mostly hexagonal assembly of the microgels at the liquid interface.

In the 4th and 5th column (Fig. 3 d,e,i,j,n,o), we zoom in onto the microgel level, focussing on the microgel morphologies. Here, it becomes evident that the microgels in flocculated emulsions have a different morphology compared to microgels stabilizing dispersed emulsions. Within the bridging points of flocculated emulsions, the microgels are adsorbed to both droplet interfaces. They are strongly deformed and form a corona on each liquid interface. On the other hand, the same microgels stabilizing dispersed emulsions are only adsorbed to one interface. There, they assume a characteristic core-corona morphology, with a swollen core extending into the aqueous phase. It is thus intuitive to assume that the difference in emulsion behaviour is linked to the substantially different microgel morphologies they assume at the liquid interfaces.

To reveal how different microgel morphologies may lead to a loss of stability above T_{VPT} , we model the stability of emulsions stabilized by microgels adsorbed to either one or two interfaces using monomer-resolved Brownian dynamics simulations. We approach both interfaces to mimic the approach of two emulsion droplets and we measure the osmotic pressure (Π) exhibited by the microgel onto each liquid interface. The simulations revealed that above T_{VPT} , emulsions stabilized by only one microgel adsorbed to two liquid interfaces induces a negative osmotic pressure onto the liquid interfaces, which explains the macroscopically observed coalescence. On the other hand, emulsions stabilized by microgels only adsorbed onto one liquid interface induce positive osmotic pressure onto the liquid interfaces, corroborating our macroscopically observed stability of the emulsions. To summarize, the combined experimental cryo-SEM study and Brownian dynamics simulations revealed that the stimuli-responsive emulsion behaviour is linked to the characteristic morphologies of the stabilizing microgels.

In summary, this paper could be suitable for publication if the main claim can convincingly be supported by experimental evidence. The latter will require the authors to revise the manuscript and make it concise and more understandable to work out the main point.

We thank Referee 3 for encouraging us to support our main claim with more experimental evidence and make our manuscript more concise. We made two major changes to the manuscript to address these points. We split figure 1 into two new figures to first discuss more concisely the impact of the new insights gained by interfacial rheology and how the new data disproves the previously proposed destabilisation mechanisms. To emphasize this point, we added to additional panels in Fig. 1 (see new figure copied above). Fig. 1a schematically illustrates the previously postulated demulsification mechanism and how they should affect the response in interfacial shear rheology. In Fig. 1b we illustrate our rheometer setup to emphasize the heating of the rheometer setup to measure changes in the interfacial properties in situ. To summarize, interfacial rheology demonstrates that the lateral

interactions within the microgel monolayers are not causing the macroscopically observed coalescence. The measurements thus hint that instead the interactions between interfaces are key.

Second, we now support our main claim that the stimuli-responsive emulsion behaviour is linked to the morphology of the stabilizing microgels with additional experimental evidence. We extended our study with two additional sets of microgels (variation in crosslinking density, core-shell towards hollow microgels) to cover the wide range tuneable physical-chemical parameters in soft microgel systems. This allows us to obtain an in-depth relationship between the architecture and morphology of the stabilizing microgels and their corresponding emulsion behaviour. We have added a summary of the extended experimental work in the key changes section above.

REVIEWERS' COMMENTS

Reviewer #1 (Remarks to the Author):

[Note from the Editor: Reviewer #1 was asked to look also over the response given to reviewer #2]

The authors mostly adequately addressed my comments and they also added significant amount of clarifying information/additional experiments. In my opinion the authors also adequately addressed the comments of the reviewer 2.

I only have the following concerns with respect to the modified version of the manuscript.

1. The plots in Fig. S1 allow one to relate the swelling ratio as a function of the temperature in experiments vs swelling ratio as a function of the model parameter “a” in simulations. Nearly order of magnitude difference in the max swelling ratio for ULC in Fig S1 1b (experiments) and 1h (modeling) needs to be addressed. Why was not lower crosslink density chosen in simulations to approach swelling ratio a bit more closely? It is typically helpful to overlap on the single plot swelling ratios in simulations with the specific experimental data the simulations are relevant to – then one can clearly see the applicability and limitations of the modeling.

2. An interesting set of experiments is added for the hydrogels with the degrading core. However, better clarification of a degradation process is needed; just the clarification of the experiments presented in this work. What does “partial degradation” mean specifically for the examples in the plots Fig. S1 (data points for partial degradation in Fig. S1 d,e,f)? What fraction of the crosslinks had degraded in these plots? The authors state that they performed systematic core degradation experiments, I however did not find how did they characterize an extent of degradation oh these hydrogels.

3. The current title is too generic, it does not really point out to specific new findings in this work. The abstract is also somewhat generic and does not fully convey the novelty of this work. The novelty of this work is however clearly conveyed in the concluding paragraphs of the Discussion section (page 12). I suggest the authors modify the title and abstract to better convey the finding of their work.

Reviewer #3 (Remarks to the Author):

I carefully read the revised version of the manuscript by Rey et al. and the rebuttal letter. The revisions made have led to significant improvements in the manuscript's quality. Notably, the issue at hand is now well described, and the authors have effectively demonstrated that the reduction in microgel size (along with decreased surface coverage of oil droplets) is not the primary cause of coalescence. This argumentation is compelling. The new scenario, supported by simulations, diverse measurements, and characterizations, is also sound and persuasive. Overall, the manuscript has been significantly improved in terms of content and readability. I think the paper is now suitable for publication.

REVIEWERS' COMMENTS

Reviewer #1 (Remarks to the Author):

[Note from the Editor: Reviewer #1 was asked to look also over the response given to reviewer #2]

The authors mostly adequately addressed my comments and they also added significant amount of clarifying information/additional experiments. In my opinion the authors also adequately addressed the comments of the reviewer 2.

I only have the following concerns with respect to the modified version of the manuscript.

We thank Referee 1 for the positive evaluation and helpful feedback.

1. The plots in Fig. S1 allow one to relate the swelling ratio as a function of the temperature in experiments vs swelling ratio as a function of the model parameter “a” in simulations. Nearly order of magnitude difference in the max swelling ratio for ULC in Fig S1 1b (experiments) and 1h (modeling) needs to be addressed. Why was not lower crosslink density chosen in simulations to approach swelling ratio a bit more closely? It is typically helpful to overlap on the single plot swelling ratios in simulations with the specific experimental data the simulations are relevant to – then one can clearly see the applicability and limitations of the modeling.

Referee 1 correctly observes that our model falls short in accurately replicating the substantial swelling ratio observed in Ultra-Low Crosslinked (ULC) microgels. It is pertinent to note that the crosslinking density of ULC microgels remains undetermined, *Scotti et al. Nature Communications 2019*. During their synthesis, we abstain from introducing a crosslinking agent, attributing the ultra-low crosslinking observed in the final particles to hydrogen abstraction facilitated by the persulfate initiator, *Virtanen et al. Soft Matter 2016*.

Despite the experimental evidence pointing towards an extremely low crosslinking density, even lower than the 0.3% parameterized in our model, we made a conscious compromise. This decision was motivated by computational constraints. Our microgels encompass a composition of 5500 monomer and crosslinker units. Drastically reducing the crosslinking density, given our finite monomer count, could yield diverse polymeric architectures that deviate from the characteristic microgel structure. For instance, in the absence of a crosslinker, we would obtain a linear polymer, while just one crosslinker could yield a 'star'-like configuration. Additionally, a sparse distribution of crosslinkers poses the risk of an asymmetric microgel due to non-uniform crosslinker dispersion. Hence, we opted for a 0.3% crosslinker density, similar to the previous work by *Bochenek et al, Nature Communications 2022*. This concentration strikes a balance, allowing for a low crosslinker density that mirrors both the ULC and 1 mol% microgels' emulsion behaviour. Importantly, it ensures that the in-silico model retains the essential features of a microgel.

The discrepancy in maximum swelling ratios for Ultra-Low Crosslinked (ULC) microgels can also be attributed to the small size of the simulated microgel. A significantly larger microgel in size amplifies entropy, inducing a stretching effect on the microgel and consequently augmenting its swelling ratio. However, due to computational constraints, our primary objective is the qualitative congruence between experimental observations and numerical

simulations. Hence, we would prefer to plot the swelling ratio from experimental microgels and in-silico microgel in separate panels, as currently displayed in Supplementary Figure 1.

We added the following sentence to the main text to clarify the choice of crosslinker concentration:

We capture the more flattened morphology of ULC microgels by decreasing their crosslinking density as well as by increasing the Wigner-Seitz cell, allowing them to spread more at the liquid interface (Figure 5i-iv, Supplementary Figure 11). We opted for a crosslinking density of 0.3%⁴⁵ due to computational constraints, striking a balance that enables a low crosslinker density while still retaining the essential features of a microgel.

In addition, we added more detailed explanation in the method section, which reads:

Although experimental evidence suggests an even lower crosslinking density for ULC microgels,⁵¹ below the 0.3% parameter used in our model, we made a considered compromise due to computational constraints. Our microgel model involves 5500 monomer and crosslinker units. Significantly reducing the crosslinking density, given our finite monomer count, could result in diverse polymeric structures that deviate from the characteristic microgel form. Additionally, a sparse distribution of crosslinkers poses the risk of creating an asymmetric microgel due to non-uniform crosslinker dispersion. Thus, we settled on a 0.3% crosslinker density, similar to the approach taken by Bochenek et al.⁴⁵ This concentration strikes a balance, allowing for a low crosslinking density that reflects the emulsion behaviour observed in both ULC and 1 mol% microgels. Importantly, it ensures that our in-silico model maintains the essential characteristics of a microgel.

2. An interesting set of experiments is added for the hydrogels with the degrading core. However, better clarification of a degradation process is needed; just the clarification of the experiments presented in this work. What does “partial degradation” mean specifically for the examples in the plots Fig. S1 (data points for partial degradation in Fig. S1 d,e,f)? What fraction of the crosslinks had degraded in these plots? The authors state that they performed systematic core degradation experiments, I however did not find how did they characterize an extent of degradation of these hydrogels.

In the previous version, we only explained the expected fraction of core degradation in the methods section, for which we apologize. We now added the expected value of 20 % core degradation in the main text, which now reads:

When partially degrading approximately 20 % of the crosslinker from the inner core,⁶⁸ the microgels become less restricted and can adapt their shape.

We also added the expected core degradation value in Supplementary Figure 1. We further removed the part ‘systematic core degradation’ as we only studied 3 microgels of the core degradation series. The respective part in the figure captions of Figure 6 and Supplementary Figure 1 now reads as.

Dodecane in water emulsions stabilized by core-shell microgels (a-e), whose inner microgel core (illustrated in purple) is either partially degraded by cleaving approximately 20 % of the crosslinks (f-j) or fully degraded (k-o) to obtain hollow microgels.

d-e) Hydrodynamic diameter D_H (d) and swelling ratio β (e) as a function of temperature for core-shell microgels, after partially degrading approximately 20 % of the crosslinker from the inner core and hollow microgels.

We should point out that we used the same synthesis protocol as reported by *Vialetto et al. ACS Nano 2021*. In their work, they performed a systematic study using static light scattering and determined and modelled the degradation rate as a function of concentration of the degrading agent and time the microgels are exposed to it. For our microgels, we expect a partial crosslinker degradation of approximately 20 % according to Vialetto et al., which is reflected in the minor change in hydrodynamic diameter (Supplementary Figure 1d) and the change in interfacial morphology observed in cryo-SEM (Figure 6g,h).

3. The current title is too generic, it does not really point out to specific new findings in this work. The abstract is also somewhat generic and does not fully convey the novelty of this work. The novelty of this work is however clearly conveyed in the concluding paragraphs of the Discussion section (page 12). I suggest the authors modify the title and abstract to better convey the finding of their work.

We agree with referee 1 and reworked the title and abstract of the manuscript to better convey our findings. With the new title we ensure that the main message of the manuscript, that interactions between interfaces rather than, as previously thought, within interfaces dictate stimuli-responsive emulsion behaviour. The new title reads as:

Interactions between interfaces dictate stimuli-responsive emulsion behaviour

We further rewrote the abstract to clearly convey the novelty of this work, which now reads as follows:

Abstract:

Stimuli-responsive emulsions offer a dual advantage, combining long-term storage with controlled release triggered by external cues such as pH or temperature changes. This study establishes that thermo-responsive emulsion behaviour is primarily determined by interactions between, rather than within, interfaces. Consequently, the stability of these emulsions is intricately tied to the nature of the stabilizing microgel particles - whether they lean towards a more polymeric or colloidal character, and the morphology they assume at the liquid interface. The colloidal properties of the microgels provide the foundation for the long-term stability of Pickering emulsions. However, limited deformability can lead to non-responsive emulsions. Conversely, the polymeric properties of the microgels enable them to spread and flatten at the liquid interface, enabling stimuli-responsive behaviour. Furthermore, microgels shared between two emulsion droplets in flocculated emulsions facilitate stimuli-responsiveness, regardless of their internal architecture. This underscores the pivotal role of microgel morphology and the forces they exert on liquid interfaces in the control and design of stimuli-responsive emulsions and interfaces.

Reviewer #3 (Remarks to the Author):

I carefully read the revised version of the manuscript by Rey et al. and the rebuttal letter. The revisions made have led to significant improvements in the manuscript's quality. Notably, the issue at hand is now well described, and the authors have effectively demonstrated that the reduction in microgel size (along with decreased surface coverage of oil droplets) is not the primary cause of coalescence. This argumentation is compelling. The new scenario, supported by simulations, diverse measurements, and characterizations, is also sound and persuasive. Overall, the manuscript has been significantly improved in terms of content and readability. I think the paper is now suitable for publication.

We thank referee 3 for the helpful feedback and the positive evaluation.